# The allotetraploid origin and asymmetrical genome evolution of the common carp *Cyprinus carpio*

Peng Xu [1,2,3,4,11]*, Jian Xu[1,11], Guangjian Liu [5,11], Lin Chen[2], Zhixiong Zhou[2], Wenzhu Peng[2], Yanliang Jiang[1], Zixia Zhao[1], Zhiying Jia[6], Yonghua Sun [7], Yidi Wu[2], Baohua Chen[2], Fei Pu [2], Jianxin Feng[8], Jing Luo[9], Jing Chai[9], Hanyuan Zhang[1], Hui Wang[2,10], Chuanju Dong [10], Wenkai Jiang [5] & Xiaowen Sun[6]

Common carp (*Cyprinus carpio*) is an allotetraploid species derived from recent whole genome duplication and provides a model to study polyploid genome evolution in vertebrates. Here, we generate three chromosome-level reference genomes of *C. carpio* and compare to related diploid Cyprinid genomes. We identify a Barbinae lineage as potential diploid progenitor of *C. carpio* and then divide the allotetraploid genome into two subgenomes marked by a distinct genome similarity to the diploid progenitor. We estimate that the two diploid progenitors diverged around 23 Mya and merged around 12.4 Mya based on the divergence rates of homoeologous genes and transposable elements in two subgenomes. No extensive gene losses are observed in either subgenome. Instead, we find gene expression bias across surveyed tissues such that subgenome B is more dominant in homoeologous expression. CG methylation in promoter regions may play an important role in altering gene expression in allotetraploid *C. carpio*.

[1] Key Laboratory of Aquatic Genomics, Ministry of Agriculture, Chinese Academy of Fishery Sciences, Fengtai, Beijing 100141, China. [2] State Key Laboratory of Marine Environmental Science, College of Ocean and Earth Sciences, Xiamen University, Xiamen 361102, China. [3] Laboratory for Marine Biology and Biotechnology, Pilot National Laboratory for Marine Science and Technology, Qingdao 266071, China. [4] State Key Laboratory of Large Yellow Croaker Breeding, Ningde Fufa Fisheries Company Limited, Ningde 352130, China. [5] Novogene Bioinformatics Institute, Beijing 100029, China. [6] Heilongjiang River Fishery Research Institute, Chinese Academy of Fishery Sciences, Harbin 150001, China. [7] Key Laboratory of Biodiversity and Conservation of Aquatic Organisms, Institute of Hydrobiology, Chinese Academy of Sciences, Wuhan 430072, China. [8] Henan Academy of Fishery Sciences, Zhengzhou 450044, China. [9] State Key Laboratory for Conservation and Utilization of Bio-Resources in Yunnan, School of Life Sciences, Center for Life Sciences, Yunnan University, Kunming 650091, China. [10] College of Fisheries, Henan Normal University, Xinxiang, Henan 453007, China. [11] These authors contributed equally: Peng Xu, Jian Xu, Guangjian Liu. *email: xupeng77@xmu.edu.cn

Two rounds of whole genome duplication (2R WGD) occurred during the evolution of early vertebrates before the divergence of lamprey from jawed vertebrates[1,2]. An additional round (3R) of whole genome duplication occurred in ray-finned fishes at the base of the teleosts. The 3R WGD, which is also known as the teleost-specific WGD (Ts3R), was estimated to happen ~320 million years ago (Mya)[3,4]. The duplication of entire genomes plays a significant role in evolution. Multiple rounds of WGD produced redundant genes, which provided an important genetic material basis for phenotypic complexity, which would potentially benefit an organism in its adaptation to environmental changes[5]. Beyond these WGD events, some teleost lineages encountered recent additional genome duplications and polyploidization. Most of the well-characterized and well-recognized polyploid fishes are in Salmonidae[6–8] and Cyprinidae[9–12] (Fig. 1). There was apparently only one auto-tetraploidization event that occurred in the common ancestor of salmonids ~100 Mya[7,8,13], while polyploidization evolved independently on multiple occasions in Cyprinids, of which the common carp (Cyprinus carpio) and goldfish (Carassius sp.) appear to have experienced the latest allotetraploidization event before their divergence[14,15], thus providing an excellent model system for investigating the initial allopolyploidization event in teleosts and understanding the evolutionary benefits for phenotypic plasticity, environmental adaptations and species radiation post the latest WGD. As one of the most important food and ornamental fishes in the Cyprinidae family, C. carpio has been widely cultured worldwide, with an annual production of over 4 million metric tons[16]. Owing to its importance in aquaculture and genome evolution studies, many efforts were made to develop genetic and genome resources in the past decades. Although the next-generation sequencing technologies and assembly

algorithms have overcome many major obstacles for whole genome sequencing and assembly, the allotetraploid nature and highly heterozygous level of C. carpio genome still bring many challenges. Polyploids usually harbor more complex genome organization and gene contents than their diploid relatives, which poses significant challenges in discriminating among homoeologous sequences and in producing high-quality genome assemblies. The first allotetraploid carp genome of its European subspecies C. carpio carpio Songpu strain had been previously finished but only anchored 52% of the scaffolds onto the 50 chromosomes[14]. The scaffolds and genes on approximately half of the genome remained ambiguous with respect to homoeologous relationships, thus creating a great obstacle for investigating the allotetraploid genome evolution of carps.

To better understand the tetraploid genome structure and gain insights into the post-WGD evolution of C. carpio, it is vital to obtain allotetraploid genomes with higher accuracy and connectivity. Here, we sequence and assemble chromosome-level allotetraploid genomes of three different C. carpio subspecies. Moreover, we also sequence four closely related diploid Cyprinid genomes and identify potential ancestral diploid lineages to discriminate between the two highly similar subgenomes. The post-WGD evolution of the newly merged allotetraploid vertebrate genome is investigated for its evolutionary history, subgenome structure and content, differentiated selective pressure, asymmetrically expressed homoeologous genes and their epigenetic regulations. Together, these resources and findings on the allotetraploid genome evolution of C. carpio provide a foundation for further unveiling the genetic basis of polyploid complexity, adaptation and phenotypic advantages and for accelerating the genetic improvement of polyploid fishes for aquaculture.

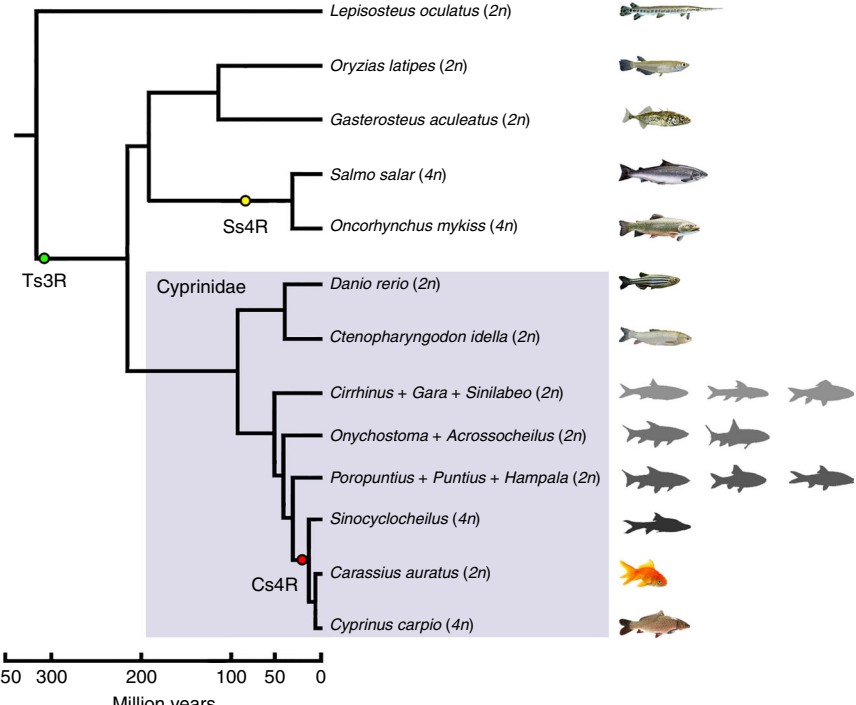

**Fig. 1** Phylogenetic relationship of tetraploid Cyprinidae and relevant teleost lineages. The phylogenetic topologies and divergence ages are taken from the TimeTree database (ref. [11]) and ref. [12]. Green, yellow and red circles represent the teleost-specific whole genome duplication (Ts3R), salmonid-specific whole genome duplication (Ss4R) and carp-specific whole genome duplication (Cs4R), respectively. The fish images are either collected and created by us (including Cyprinus carpio, Ctenopharyngodon idella, Oryzias latipes, Danio rerio, Cirrhinus, Garra, Sinilabeo, Onychostoma, Acrossocheilus, Poropuntius, Puntius, Hampala and Sinocyclocheilus) or obtained from public domains (including Lepisosteus oculatus, Gasterosteus aculeatus, Salmo salar, Oncorhynchus mykiss and Carassius auratus)

## Results and discussion

**Genome assembly, annotation and scaffold anchoring.** We sequenced the tetraploid genomes of three distinct *C. carpio* strains, namely, Hebao red carp (HB) and Yellow River carp (YR) from China, which belong to the subspecies *C. carpio haematopterus*, and German mirror carp (GM) from Europe, which belongs to the subspecies *C. carpio carpio*, by whole-genome shotgun methods (Supplementary Fig. 1, Supplementary Table 1). The three assembled genomes spanned 1460 Mb, 1425 Mb, and 1416 Mb, with the contig N50 and scaffold N50 of 20.68 kb and 923.37 kb for HB, 21.81 kb and 1706 kb for YR, and 52.14 kb and 3466 kb for GM, respectively (Supplementary Table 2). We constructed three high-resolution genetic maps with 29,019 (HB), 28,194 (YR) and 32,160 (GM) markers on 50 chromosomes by genotyping the mapping families using a carp 250 K SNP array[17] (Supplementary Fig. 2, Supplementary Tables 3). We then anchored the assembled genomes to the 50 chromosome frames of the genetic maps. Finally, three chromosome-level reference genomes of *C. carpio* were created with high connectivity, representing 1.24 Gb (82%) of HB, 1.26 Gb (89%) of YR and 1.3 Gb (92%) of GM assemblies, respectively (Supplementary Table 4). All three assemblies represent a substantial improvement over the previously published draft genome sequence of *C. carpio*, which only anchored 875 Mb (52%) onto the 50 chromosomes[14].

We BLAST-aligned highly conserved core eukaryotic genes (Cluster of Essential Genes (CEG) database) to the genome assemblies with a core eukaryotic genes mapping approach (CEGMA) pipeline, which showed high-confidence hits identified in three assembled genomes of *C. carpio* (Supplementary Table 5). We also validated the assembled genomes by matching them with expressed sequence tags (ESTs) downloaded from the US National Center for Biotechnology Information (NCBI) database, which indicated that 99.86%, 99.26%, and 99.20% of the ESTs were covered by the assembled genomes of HB, YR, and GM, respectively (Supplementary Table 6). We further compared the assembled genome of GM with a previously published draft genome of the mirror carp Songpu strain (SP). Both the GM and SP belong to the European subspecies *C. carpio carpio*. To assess the genome connectivity and assembly accuracy, we aligned 34,932 mate-paired BAC-end sequences (BES) that were derived from the SP genome to both SP and GM assemblies. The result showed that 98% and 97% of the BESs were mapped to the genomes of SP and GM, respectively. In over 85% of BES mate-pairs, both BESs were aligned to the same scaffold of GM, compared to only 31% of the BES mate-pairs that aligned to the scaffold of the SP draft genome. We observed the standard Poisson distribution of the BES pair intervals on GM scaffolds that corresponded to the real BAC insertion length, while no regular distribution was observed on the SP scaffolds (Supplementary Fig. 3). A comparative analysis of three *C. carpio* genome assemblies revealed that the three genomes are highly conserved at the chromosome level, with only a limited number of large structural or segmental variations (Supplementary Fig. 4). We identified 557.87 Mb of repetitive sequences from the HB genome, 518.98 Mb from the YR genome and 494.10 Mb from the GM genome, which contributed to 36.94%, 36.42%, and 34.89% of three genomes, respectively, with similar classifications and proportions (Supplementary Table 7). The most abundant transposable elements were DNA transposons, which contributed to ~13% of all three genomes, with Tc1 mariners representing ~5% of the genomes. We annotated 44,269, 44,626 and 44,758 protein-coding genes in the HB, YR, and GM genomes, respectively (Supplementary Table 8 and 9). Approximately 96.9%, 96.2% and 96.2% of HB, YR and GM genes could be annotated by non-redundant nucleotides and proteins in the SWISS-PROT, Gene Ontology (GO), Kyoto Encyclopedia of Genes and Genomes (KEGG), Cluster of Orthologous Groups (COG), Pfam and NCBI databases (Supplementary Table 10). Together, the evidence suggested that the three newly assembled genomes of *C. carpio* had been improved significantly in connectivity and contiguity, thus paving the way for further unveiling of the allotetraploid genome evolution of Cyprinids.

**Allotetraploid origin of the common carp *C. carpio*.** The common carp *C. carpio* resulted from the ancient hybridization of two ancestral diploid cyprinid species[9], which is of critical importance for genome evolution studies that divide the allotetraploid genome into two subgenomes, thereby representing two ancestral diploid genomes. Cyprinids are a diverse teleost family with over 2400 valid species in at least 220 genera[18,19], of which the subfamily Cyprininae comprises over 1300 species in four groups of barbine, cyprinine, labeonine and schizothoracine with diversified and complex karyotypes from $2n = 50$ to ~470[12]. Under such circumstances, it is very challenging to identify diploid ancestral lineages and unveil the evolutionary origin of allotetraploid common carp. Previously, we successfully recognized 25 homoeologous chromosome pairs in the *C. carpio* genome by aligning 50 chromosomes of *C. carpio* into 25 chromosomes of the zebrafish (*Danio rerio*) diploid genome[14,20,21], thereby revealing the existence of two homoeologous sets of chromosomes in the *C. carpio* genome. To explore the evolutionary relationship of *C. carpio* and its closely related tetraploid and diploid Cyprininae species, we constructed a phylogenetic tree of the representative Cyprininae species using a nuclear gene, *recombination activating gene 2* (*rag2*), which presents only one copy in diploid cyprinids but two copies in tetraploid cyprinids (Fig. 2a, Supplementary Fig. 5, Supplementary Table 11)[12,22]. The phylogenetic topology revealed that two copies of the *rag2* of the closely related tetraploid genomes, including three *C. carpio*, three cavefish *Sinocyclocheilus* and goldfish *Carassius auratus* genomes, were clustered into two distinct homoeologous clades, which suggested that these tetraploid cyprinids either derived from a common tetraploidization event and then diverged into different species or experienced independent and recurrent tetraploidization events involving the same or closely related diploid ancestors. One of the homoeologous clades of *rag2b* from the tetraploid genomes subsequently joined with *rag2* genes from a diploid group in Barbinae, including multiple genera, such as *Poropuntius*, *Puntius*, *Hampala* and *Onychostoma*, which implied that one of the two diploid progenitors of *C. carpio* may have derived from a diploid Barbinae species. However, the *rag2a* clade did not merge with any *rag2* genes from the diploid species. The phylogenetic results gave us clues on how to separate the two subgenomes of the allotetraploid *C. carpio* genome. We reasoned that the genome similarity and alignment coverage between the diploid progenitors and descendent allotetraploid on each homoeologous chromosome pair would facilitate the discrimination of two descendent subgenomes. We thus selected three species from Barbinae (*Poropuntius huangchuchieni*, *Hampala macrolepidota* and *Onychostoma barbatulum*) as progenitor-like diploid candidates and one species from a relatively distant lineage (*Cirrhinus molitorella*) as a reference diploid species for whole genome sequencing and draft assembly (Supplementary Fig. 6, Supplementary Table 12). The genome sequences of the four newly sequenced diploid Cyprinids and two previously sequenced diploid Cyprinids (*Ctenopharyngodon idella* and *D. rerio*) were then aligned to 25 pairs of homoeologous chromosomes of *C. carpio* genomes. The results showed that the genome sequences of three diploid Barbinae species have an extraordinarily higher similarity and coverage in one chromosome than in the other for each homoeologous chromosome pairs

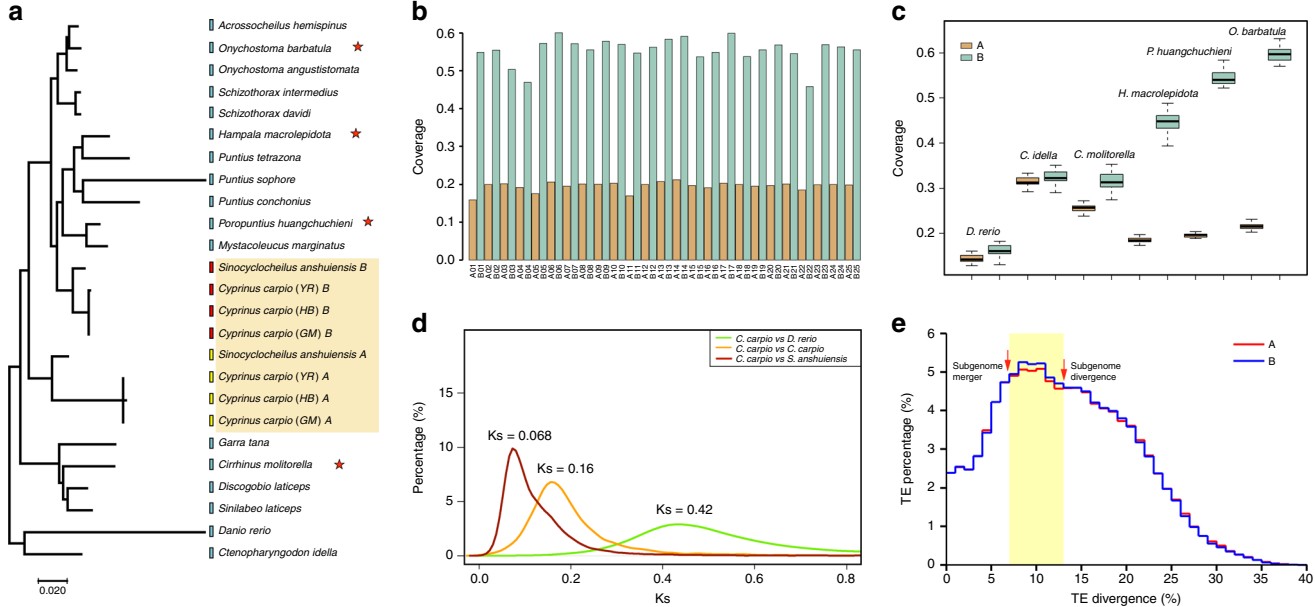

**Fig. 2** Allotetraploid origin and evolution history of *C. carpio*. **a** Phylogenetic relationship of *rag2* orthologues of *C. carpio* and its tetraploid and diploid close relatives in the subfamily Cyprininae. The pentagrams indicate three selected diploid species (*Poropuntius huangchuchieni*, *Hampala macrolepidota*, and *Onychostoma barbatulum*) as progenitor-like diploid candidates, and one species from a relatively distant lineage (*Cirrhinus molitorella*) for genome sequencing to represent the closely related diploid lineages from Cyprininae. **b** A histogram shows the coverage of *P. huangchuchieni* genome sequence mapping to 50 chromosomes of *C. carpio*. Chromosome IDs have been re-assigned to represent two sets of homoeologous chromosomes. **c** Boxplots show the genome coverage and similarity comparisons of diploid relatives to the tetraploid genome of *C. carpio*. **d** The distribution of the synonymous substitution rates (Ks) of homologous genes between *D. rerio* and *C. carpio*, *C. carpio* and *Sinocyclocheilus*, and homoeologous genes between two subgenomes of *C. carpio*. Three peaks (Ks = 0.42, 0.16 and 0.068) of Ks distribution indicate the divergences of *D. rerio* and *C. carpio*, *C. carpio* and *Sinocyclocheilus*, and two progenitors of *C. carpio*. **e** The distribution of sequence divergence rates of transposable elements (TEs) as percentages of subgenome sizes of *C. carpio*. The TE content segregation between subgenomes A and B indicates the events of diploid progenitor divergence and subgenome merger

without exception, while no significant similarity or coverage differences were observed when the genome sequences of *C. molitorella*, *C. idella* or *D. rerio* were aligned to the homoeologous chromosome pairs of *C. carpio* (Fig. 2b, c). The results provided substantial evidence of the allotetraploid origin of *C. carpio* and suggested that one subgenome progenitor possibly originated from a diploid lineage of Barbinae, while the other progenitor might be an unexplored or even extinct diploid from a relatively distant lineage in Cyprinids. We therefore divided 50 chromosomes evenly into subgenomes A and B, which represented the unknown progenitor A and the ancient progenitor B from Barbinae, respectively. We further performed a phylogenomic analysis based on 2071 conserved homoeologous gene pairs from two allotetraploids (*C. carpio* and *S. anshuiensis*) and their single-copy orthologs from three diploids (*D. rerio*, *C. idella*, and *P. huangchuchieni*). The conserved phylogenic topology confirmed the homoeologous pattern of two subgenomes in *C. carpio* (Supplementary Fig. 7), further supporting our hypothesis that subgenome B of *C. carpio* originated from a diploid Barbinae species. These findings built the foundation for studying the allotetraploid origination and genome evolution of *C. carpio* and its closely related species.

To estimate the accurate time of the Cyprinid-specific allotetraploidization event, we calculated the synonymous substitution rates (Ks) of 8270 homoeologous genes to determine the divergence time of the two subgenomes. These homoeologous genes present only one copy in the diploid genome of *D. rerio* and one copy in each of the two subgenomes in the allotetraploid genome of *C. carpio*. The substitution rates of *Danio-Cyprinus* orthologous genes, the homoeologous genes in subgenomes A and B in *C. carpio*, and the *Cyprinus-Sinocyclocheilus* orthologous genes were calculated to be 0.42, 0.16, and 0.068, respectively

(Fig. 2d). We applied the previously determined molecular clock that Ks in teleost was ~$3.51 \times 10^{-9}$ substitutions per synonymous site per year[9] and presumed the evolutionary rates were consistent in two subgenomes. We therefore estimated that *D. rerio* and *C. carpio* diverged ~60 million years ago (Mya), the two ancient progenitor species of *C. carpio* diverged ~23 Mya, and *Sinocyclocheilus* and *Cyprinus* diverged ~9.7 Mya. Thus, we estimated that the Cyprinid-specific WGD and allotetraploidization event most likely occurred after the divergence of two ancient progenitors (23 Mya) but before the divergence of the two allotetraploid lineages of *Sinocyclocheilus* and *Cyprinus* (9.7 Mya). We further collected transposable elements (TE) from two subgenomes and assessed their divergence rates in each subgenome (Fig. 2e and Supplementary Fig. 8). The result showed that TE sequence divergence between two subgenomes displays high degree of overlap, suggesting the consistency of TE evolutionary rate between two subgenomes. Intriguingly, we identified differentiated TE contents in subgenomes A and B with divergence rates from 7% to 13%, which formed a "bubble" peak in the TE divergence profile (Fig. 2e). This suggested that TE substitution rates in subgenomes A and B differentiated after their divergence into two independent diploids at ~23 Mya and were re-unified in the allotetraploid genome after the Cc4R event. We therefore estimated that the WGD event likely occurred ~12.4 Mya based on the TE substitution rate at the re-unified point, which was consistent with previous estimations based on fossil[23] and molecular evidences[9], but much earlier than the estimation based on the previous draft genome[14].

**Subgenome structure and gene content**. We divided the allotetraploid *C. carpio* genome evenly into two distinct subgenomes,

A and B, with 25 chromosomes in each based on the differentiated sequence similarities in comparison with the genome of progenitor B from Barbinae. The new chromosome IDs were then assigned to 50 chromosomes based on their homologous relationship with 25 chromosomes of *D. rerio* (Supplementary Table 13). For instance, chromosomes A01 and B01 were assigned to a pair of homoeologous chromosomes that are syntenically related to chromosome 1 of *D. rerio*, of which A01 belongs to subgenome A and B01 belongs to subgenome B. The new chromosome IDs will facilitate better understanding of the homoeologous landscape of the allotetraploid *C. carpio* genome.

Previous studies on allopolyploid genomes, mostly in plants, revealed that one of the parental subgenomes often retains significantly more genes and exhibits significantly higher expression, stronger purifying selection and a lower DNA methylation level than those of the other subgenome. This phenomenon was referred to as subgenome dominance[24,25]. It is essential to investigate the homoeologous gene contents, the expression profile and their epigenetic regulations for verifying subgenome dominance and to better understand the allotetraploid genome evolution after genome merger. We assigned 21,078 genes in 633 Mb sequences to 25 chromosomes of the subgenome A and 22,099 genes in 671 Mb sequences to 25 chromosomes of the subgenome B, indicating that the gene content of subgenome B is slightly higher than that of subgenome A (Supplementary Data 1). The GC content, gene structure, and repetitive element distribution did not show significant differences between the two subgenomes (Supplementary Fig. 8, Supplementary Data 1). To assess the fate of the homoeologous genes in the *C. carpio* genome, we compared the gene contents of the allotetraploid genome of *C. carpio* and the diploid genome of *C. idella*, which is the most closely related diploid Cyprinid with a completely sequenced genome available for serving as the diploid orthologous reference. We built a total of 10,724 orthologous gene pairs or triplets within the two genomes, including 8291 orthologous gene triplets that presented one copy in the diploid *C. idella* genome and one copy in each of two homoeologous chromosomes in two subgenomes of the *C. carpio* genome. Visualization of the chromosomal locations of 8291 homoeologous gene pairs revealed a high degree of chromosome-level synteny between the subgenomes A and B (Fig. 3a, Supplementary Fig. 4). We also identified 915 and 1220 single-copy orthologous genes in the subgenomes A and B of *C. carpio*, respectively, accounting only a small portion of the gene contents in two subgenomes (Supplementary Table 14). A gene ontology (GO) analysis indicated that gene loss was not random in the *C. carpio* genome after the latest WGD. Single-copy genes are overrepresented in some essential functional categories in both subgenomes, such as the nucleic acid metabolic process (GO:0090304), DNA repair (GO:0006281), DNA replication (GO:0006260), nuclease activity (GO:0004518) and ribonucleoprotein complex (GO:1990904), which was consistent with previous studies in various tetraploid genomes[26–29] (Supplementary Data 2) and suggested that these genes might be sensitive to altered gene dosage during WGD and might lose reciprocally to return single-copy status in two subgenomes. We found that only a small portion of homoeologous gene pairs (118 homoeologs) were mapped within the same subgenome, suggesting that limited homoeologous exchange events have occurred across two subgenomes after the WGD event (Supplementary Data 3). Besides, we found a total of 92 large segmental rearrangements, including 40 segmental inversions, between the homoeologous chromosomes of two subgenomes. For example, we found a segmental inversion of 2.5 Mb and a segmental translocation of 1.8 Mb presenting in the homoeologous chromosomes A24/B24 and A15/B15 of the *C. carpio* genome, respectively (Supplementary Fig. 9). We

validated 35 segmental homoeologous rearrangements by mapping mate-paired BESs spanning the breakpoints of the homoeologous chromosomes (Supplementary Table 15 and Supplementary Data 4). The chromosome-level TE distribution analysis revealed that TEs are significantly enriched in the flanking regions of the structure variations between the two subgenomes, suggesting that TEs could be the potential driving force of those homoeologous exchanges and segmental variations in the allotetraploid genome of *C. carpio*. Overall, the gene content and syntenic analyses suggested that two homoeologous gene sets were well preserved in the two subgenomes. We did not observe extensive gene losses and rediploidization in the allotetraploid genome of *C. carpio*.

**Subgenome expression bias in the allopolyploid *C. carpio* genome.** Subgenome dominance usually leads to stronger purifying selection and the dominant gene expression of homoeologous genes in many tetraploid genomes of plants and animals[13,25,30,31]. To assess the selective pressure of two subgenomes, we calculated both the nonsynonymous substitution rate (Ka) and Ks values based on homoeologous gene pairs (Supplementary Data 5). Intriguingly, the results showed that all 25 chromosomes in subgenome A had a significantly higher Ka/Ks ratio (mean Ka/Ks = 0.20) than their homoeologous chromosomes in subgenome B (mean Ka/Ks = 0.18) ($p = 1.28e-15$). The distinct selective pressure difference of 25 paired homoeologous chromosomes indicated asymmetric evolution of two subgenomes in *C. carpio*. The subgenome A genes are under relaxed purifying selection and are evolving faster than their homoeologs in subgenome B. The subgenome B genes are under greater purifying selection than their homoeologs in subgenome A (Fig. 3b). A previous study on the tetraploid maize genome suggested that dominant gene expression is a strong determinant of the strength of purifying selection[32]. To investigate whether the similar phenomena also presents in the *C. carpio* genome, we therefore compared the genome-wide transcriptional levels of subgenomes A and B based on the gene expression levels of 8291 homoeologous gene pairs in 12 tissues to investigate the homoeologous gene expression patterns and their divergence in two subgenomes (Fig. 3c, Supplementary Table 16, Supplementary Data 6 and 7). The results indicated that a total of 7536 expressed homoeologous gene pairs (~91%) had expression differences greater than 2-fold change in at least one tissue, including 4719 and 5403 homoeologs having higher expression in subgenomes A and B, respectively (Supplementary Data 8), of which, 2133 and 2817 homoeologs had higher expression values exclusively in the 12 tissues of subgenomes A and B, respectively, while 2586 homoeologs had swinging expression bias in 12 tissues. The homoeologous expression bias showed asymmetric expression patterns between subgenomes A and B so that the genome-wide expression level dominance was biased towards subgenome B in the allotetraploid *C. carpio* genome (Supplementary Table 17). We tuned the threshold of expression change to identify those homoeologs with extraordinary expression divergence in the two subgenomes. We found that 1018 homoeologous genes had expression differences greater than 32-fold between the two subgenomes, including 406 and 627 homoeologs that had higher expression exclusively in subgenomes A and B, respectively. Gene annotation indicated that the dominantly expressed homoeologs of subgenome A were enriched in "nucleobase biosynthetic and metabolic processes", "lipid metabolic process" and "lipid biosynthetic process", while dominantly expressed homoeologs of subgenome B were enriched in "oxidoreductase activity", "hydrolase activity", "response to stress" and "DNA repair" (Supplementary Data 9).

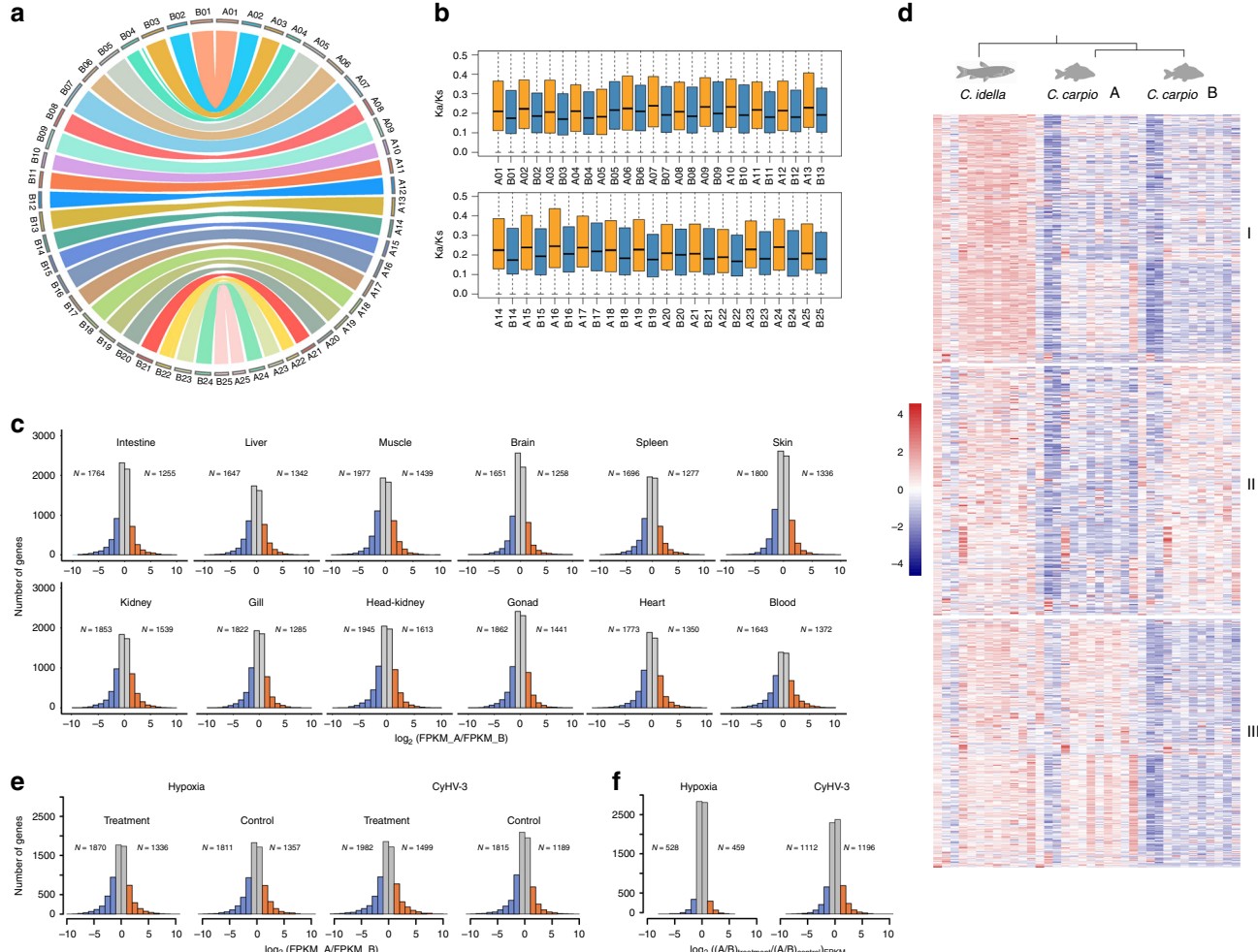

**Fig. 3** Asymmetrical homoeologous gene expression in *C. carpio*. **a** Circos plot distribution of homoeologous gene pairs in 25 chromosome pairs across subgenomes A and B. **b** Boxplot of the Ka/Ks ratio distribution of protein-coding genes in 50 chromosomes of *C. carpio*. Orange and blue boxplots indicate that the chromosomes belong to subgenomes A and B, respectively. **c** Histograms of the genome-wide expression of homoeologous genes among the indicated tissues of *C. carpio*. Log$_2$(FPKM_A/FPKM_B) indicates the degree of expression difference of homoeologous gene pairs. N values indicate the number of dominant genes in subgenomes A and B. **d** Heatmaps of three divergently expressed triplet clusters (each triplet includes two homoeologous genes of *C. carpio* and their orthologue of *C. idella*), indicate potential subfunctionalization and neofunctionalization in the allotetraploid genome of *C. carpio*. The orders of 12 tissues in both *C. idella* and *C. carpio* are the same as 3c. **e** Histograms of the genome-wide expression divergence of homoeologous genes of *C. carpio* in stress treatments and controls. N values indicate the number of dominant genes in subgenomes A and B, respectively. **f** Histogram of the ratio of homoeologous expression divergence in the stress treatment and controls [(A/B)$_{treatment}$/(A/B)$_{control}$], which indicates accelerated homoeologous expression divergence in stress treatments compared to controls

We further built eight co-expression clusters of all expressed genes across all 12 tissues to assess the overall expression divergence rate between the two subgenomes (Supplementary Fig. 10). We investigated the 8291 homoeologs in eight co-expression clusters and found that 1986 pairs of homoeologous genes (~24%) had been assigned to different co-expression clusters. The results suggested that substantial spatial expression partitioning of the differentially expressed homoeologous genes occurred, while the majority of the homoeologous genes still tend to retain similar expression levels and patterns in the allotetraploid genome of *C. carpio*. The spatially differentially expressed homoeologous gene pairs in two subgenomes may have experienced or been experiencing functional divergence via the subfunctionalization or neofunctionalization mechanism, which are commonly observed in allopolyploid genomes after genome mergers and gene duplications. To discriminate which functional divergence mechanism potentially occurred on specific homoeologs, we further collected transcriptomic data from the

same 12 tissues of *C. idella* (Supplementary Table 18). We built co-expression clusters based on 8214 expressed orthologous triplet genes in *C. idella* and *C. carpio*. The expression patterns of 306 orthologous triplets were differentially expressed in *C. idella* and in two subgenomes of *C. carpio*, suggesting that the subfunctionalization mechanism potentially re-shaped the spatial expression patterns of these homoeologs of *C. carpio*. We further identified that 293 genes in subgenome A and 228 genes in subgenome B had conserved co-expression patterns with their orthologues of *C. idella*, respectively, while their homoeologous copies in the opposite subgenome were differentially expressed, suggesting that neofunctionalization mechanism potentially occurred in one of the copies (Fig. 3d). Majority of the genes in three sub-clusters were included in 1986 divergent homoeologous genes (Supplementary Fig. 11, Supplementary Data 10). In addition, we also built co-expression clusters based on expressed homoeologous gene pairs of *C. carpio* without diploid outgroup (Supplementary Fig. 12). We found that 191 and 620

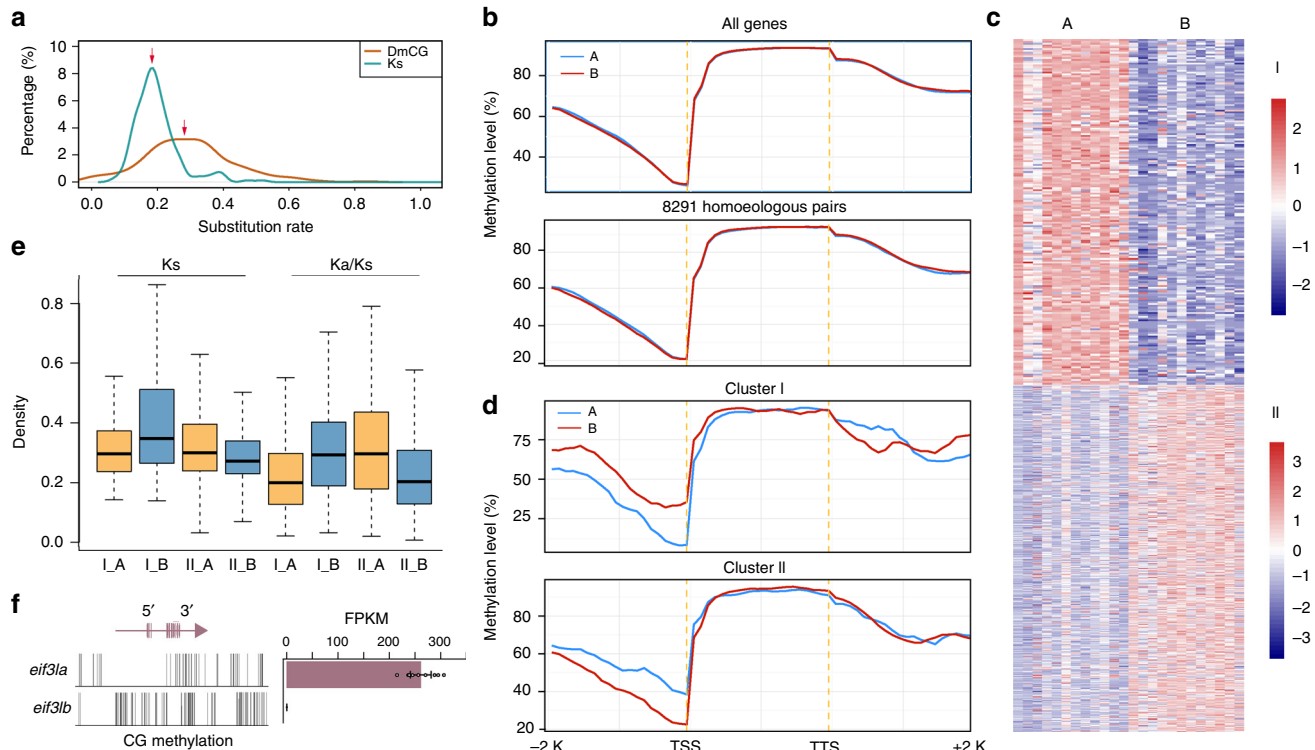

**Fig. 4** DNA methylation in homoeologous expression bias in *C. carpio*. **a** Distribution of synonymous substitution values (Ks) (blue) and gene-body DmCG percentages (red) of 2393 methylated homoeologous genes. Peak values are indicated by arrows. **b** CG methylation levels of all annotated genes and 8291 homoeologous gene pairs in two subgenomes. **c** Heatmaps of two extremely divergent co-expression clusters, of which one of two homoeologous genes in subgenomes A or B was extensively transcribed while the other copies suppressed in 12 tissues. **d** CG methylation levels of divergent expressed homoeologous genes corresponding to clusters I and II in **c**. **e** Boxplot of the Ka/Ks ratio distribution of homoeologous genes of two extremely divergent expression clusters in two subgenomes as shown in **c**. Yellow and blue boxplots indicate the genes in subgenomes A or B, respectively. **f** CG methylation level and expression level of homoeologous genes *eif3l* of *C. carpio*, which demonstrated that reduced CG methylation levels in the promotor region correlated with increased expression levels of *eif3la*, while *eif3lb* was heavily methylated and silenced. FPKM reads per kilobases per million. Data represent mean ± sem for *n* = 9 expressed tissues

homoeologous gene in subgenomes A or B were extensively transcribed in 12 tissues, respectively, while the other copies were barely transcribed in 12 tissues, indicating that nonfunctionalization may have occurred and suppressed one copy of the homoeologous pairs (Fig. 4c, Supplementary Data 11). Selection pressure on these 811 homoeologs with an extreme divergent expression in two subgenomes was then investigated, which showed that the dominantly transcribed homoeologous copies, regardless of their subgenome location, were more likely experiencing stronger purifying selection than their homoeologs (Fig. 4e). The observed connection of strength of purifying selection and expression dominance was consistent with previous findings in polyploid plants[25]. We further investigated the gene functions of these functionally divergent genes or silenced genes in either subgenome A or B and found that some vital functional categories of GO and KEGG were disproportionately over-represented, including "fatty acid metabolism (KO01212)", "RNA degradation (KO03018)", "ribosome biogenesis (KO03008)", "nucleotide excision repair (KO03420)", "peroxisome (KO04146)", "lipid metabolic process (GO:0006629)", "nucleic acid metabolic process (GO:0090304)", and "oxidation-reduction process (GO:0055114)", which are potentially involved in critical pathways and require optimal stoichiometry of gene expressions to ensure normal biological processes and the survival of tetraploid *C. carpio* after genome merging (Supplementary Data 12). For example, we found that the eukaryotic translation initiation factor 3 subunit I (*eif3I*) gene located on chromosome A19 was dominantly expressed in all 12 surveyed tissues, while its

homoeologous copy on chromosome B19 was completely silenced (Fig. 4f). eIF3 is a multiprotein complex that functions during the initiation phase of eukaryotic translation[33]. Altered expression levels of eIF3 subunits correlate with neurodegenerative disorders and cancer development and may also trigger infection cascades[34]. We also found that the expression of the programmed cell death 1 ligand 1 (*pd-l1*) gene on chromosome B13 was suppressed in all 12 tissues, while its homoeolog on chromosome A13 maintains normal expression (Supplementary Fig. 13). The Pd-l1 gene had near-ubiquitous expression in tissues and played a major role in suppressing the immune system by dampening the T cell response and preventing overactivation during proinflammatory states[35,36]. Similarly, the long-chain fatty acyl-CoA ligase 6 (*acsl6*) gene in chromosome B19 of subgenome B was silenced when its homoeolog on chromosome A19 of subgenome A was dominantly expressed (Supplementary Fig. 13). The Acsl6 gene was mainly expressed in neural cells and the brain that was essential for regulating the partitioning of acyl-CoA species towards different metabolic fates, such as lipid synthesis or β-oxidation[37]. These genes were likely involved in dosage-sensitive regulation pathways by which the suppressed expression of one homoeologous copy would ensure optimal RNA and protein supplies and maintain normal biological processes.

Polyploidization may confer a significant adaptive advantage in response to various environmental challenges during the evolution history[38]. To explore whether homoeologous gene expression bias plays a role in stress responses, we collected transcriptome expression data from *C. carpio* under hypoxia and disease stresses

with Cyprinid herpesvirus 3 (CyHV-3) and the bacterial pathogen *Aeromonas hydrophila* separately, as well as data from control samples from the same experiments. We investigated homoeologous expression divergence in 8291 previously identified homoeologous gene pairs in both stress treatments and controls. We still observed asymmetric homoeologous expression patterns in all surveyed samples of biotic or abiotic stress treatments and their controls and found that subgenome B was still dominantly expressed (Fig. 3e, Supplementary Fig. 14, Supplementary Table 19). We also found that samples under stress treatment tend to have more homoeologs that are divergently expressed in two subgenomes than the control samples (Fig. 3f). We performed differential gene expression (DEG) analysis of the homoeologous genes and identified 3367 (23.47%), 5768 (38.70%), and 3448 (23.67%) genes that were differentially expressed with 2-fold changes compared with corresponding controls in the stress experiments with hypoxia, CyHV-3 and *A. hydrophila* stresses, respectively (Supplementary Table 20). To evaluate the homoeologous expression responses to specific stress treatments within the allotetraploid genome, we calculated the summed expressions of homoeologous genes in both treatments and controls, which are hereafter referred to as Treatment $_{(A+B)}$ and Control $_{(A+B)}$ expression. The results indicated that 1155 (17.22%), 2297 (32.56%), and 1187 (17.45%) homoeologous pairs had Treatment $_{(A+B)}$ expression values 2-fold greater than Control $_{(A+B)}$ expression during three stress treatments (Supplementary Table 21), which were much less than the differentially expressed genes that were identified in traditional DEG analysis. The WGD increases gene redundancy and increases the expression plasticity of the allotetraploid genome, thus buffering gene expression in response to environmental stresses. The genes with significant expression changes could be buffered by the expression of its homoeologous copy and reduce the drastic impact of expression changes. Considering that the majority of homoeologous genes still maintain conserved expression patterns and gene functions, the DEG analysis based on summed expressions of homoeologous genes would provide an alternative approach for investigating gene regulations underlying phenotypic variation.

Taken together, the results of our expression divergence analysis using different methods provide substantial evidence of subgenome dominance and asymmetric expression, of which subgenome B not only retains more protein-coding genes but also has dominant gene expression levels under a stronger purifying selection in the allotetraploid genome. However, it is still a challenging task to identify and discriminate among those mechanisms of gene functional divergence[39,40], especially when accurate duplicate-aware gene annotation, well-powered transcriptomic analysis and expression studies, including embryonic stages, are absent for both allotetraploid *C. carpio* and the closely related diploids from its progenitor lineages.

**Methylation divergence in allopolyploid *C. carpio* genome.** It is well recognized that DNA methylation is one of the important epigenetic regulation mechanisms for controlling gene expression[41]. To uncover DNA methylation divergence underlying expression divergence between two subgenomes, we generated single-base resolution methylomes from the YR strain of allotetraploid *C. carpio* (Supplementary Table 22). We identified 309,953,955 conserved cytosines with approximate 21,724,824 cytosines methylated (7.01%) present in three biological replicates for further analysis. We found that the cytosine methylation rates in subgenomes A and B were 6.88% and 7.13%, respectively, with no significant difference. The most abundant methylation occurs at CG sites in both subgenomes, with a CG methylation ratio of

85.98% in subgenome A and 86.34% in subgenome B, while the CHG and CHH methylation ratios are much lower (Supplementary Table 23). Therefore, we decided to focus on CG methylation in the methylation divergence analysis across two subgenomes. To test the relationship between DNA methylation changes and sequence evolution in genic regions, we identified CG body-methylated genes in the 8291 homoeologous gene pairs of two subgenomes using a binomial test with body-methylation levels[42,43]. Among the 2393 CG body-methylated homoeologous genes, the percentage of CG methylation changes was substantially higher than the substitution rate of the coding sequence (Fig. 4a), which suggested that the accumulated methylation change rate was faster than the neutral sequence substitution rate since two subgenomes diverged. The faster methylation change rate suggested that, rather than vast gene loss and sequence substitution, epigenomic evolution provided quick and efficient regulation to deal with the massive changes of "genome shock" after the genome merger and ensured the survival of the tetraploidized *C. carpio*.

Many evolutionary studies on allotetraploid genomes suggested that dominantly expressed subgenomes tend to have less methylation sites in genic regions[42,44,45]. Therefore, we compared the global DNA methylation level of two subgenomes and looked for epigenetic evidence of subgenome dominance in *C. carpio* based on all annotated genes. We found that two subgenomes have similar CG methylation levels in the gene body and downstream regions, while subgenome A has a slightly higher CG methylation level in the upstream promotor regions than that of subgenome B (Student's *t*-test, $p < 0.05$). We found a similar result when we compared the CG methylation of the highly-conserved 8291 homoeologous gene pairs between two subgenomes (Fig. 4b). The results suggested that the two subgenomes that were derived from different diploid progenitors retain a balanced CG methylation level in the gene body and downstream regions after the genome merger, which are not likely to be responsible for subgenome dominance and asymmetric expression in the allotetraploid genome. We reasoned that methylation changes in the promotor regions of the homoeologous gene pairs potentially regulate homoeologous expression bias and lead to subgenome dominance. Therefore, we further investigated CG methylation patterns among those that were divergently expressed homoeologous genes with an expression difference greater than 32-fold in 12 surveyed tissues. We found that the CG methylation levels in the promoter regions are much lower in the more highly expressed homoeologs than in the poorly expressed homoeologs in all 12 tissues, regardless of whether the higher expression homoeologs are in subgenome A or subgenome B. However, we did not observe a significant CG methylation difference in the gene body and downstream region between these asymmetrically expressed homoeologous gene pairs (Supplementary Fig. 15). We further investigated the methylation divergence patterns of the homoeologous genes with extreme expression divergence, which are only expressed in one subgenome while they are suppressed in the other subgenome (Fig. 4c). The result indicated that the genes that were dominantly expressed in subgenome A retain a significantly lower level of CG methylation in the promoter region than their suppressed homoeologs in subgenome B ($p = 2.06e-07$). Similarly, the dominantly expressed homoeologs in subgenome B also have a significantly reduced CG methylation level in the promoter region compared with their homoeologs in subgenome A ($p = 4.23e-08$). However, we did not find significant methylation divergence in the gene body or the downstream regions (Fig. 4d, Supplementary Fig. 16). As expected, the silenced *eif3I* as a typical example, has significantly increased CG methylation levels in the promoter regions than their normally expressed homoeologs (Fig. 4f). Thus, we found a

significant correlation between asymmetric expression and methylation divergence in the promoter region, which underpinned that CG methylation in the promoter regions plays an important role in altering the expression of these homoeologous genes in allotetraploid *C. carpio*. We suggest that selective methylation changes in homoeologous genes fine-tuned the gene expression to generate proper transcriptional products, which ensures the survival and evolutionary success of the tetraploidized *C. carpio*.

## Methods

**Ethics statement**. This study was carried out in accordance with the recommendations of the care and use of animals for scientific purposes set up by the Animal Care and Use Committee of Chinese Academy of Fishery Sciences (ACUC-CAFS). The protocol was approved by the ACUC-CAFS. The fish was euthanized in MS222 solution before samples were collected.

**Sample preparation, genome sequencing and assembly of common carp**. We collected a female Yellow River carp (*C. carpio haematopterus*) at the Hatchery Station of Henan Academy of Fishery Sciences at Zhengzhou; a female Hebao red carp (*C. carpio wuyuanensis*) at Wuyuan County in Jiangxi Province, China; and a female German Mirror carp (*C. carpio carpio*) at Heilongjiang Fishery Research Institute in Harbin as genomic DNA donors for whole genome sequencing. Genomic DNA was extracted from blood using a QIAGEN DNeasy Blood & Tissue Kit (QIAGEN, Shanghai, China). We constructed eight sequencing libraries for three carps with various insert sizes from 250 bp to 20 Kbp according to Illumina standard operating procedures. Subsequently, we used the Illumina HiSeq platform to sequence these libraries with 150-bp read length, and generated 339.11 Gb, 298.65 Gb, and 330.39 Gb clean data for three genome assemblies. We estimated the genome size based on 17-mer frequency distribution and assembled the genome via a modified version of SOAP*denovo*, specifically for the high heterozygous genome[46]. To evaluate the accuracy of the assemblies at single base level, we mapped short sequence reads to carp genomes with BWA[47] and performed variant calling with SAMtools[48]. We also assessed assembly completeness by remapping the PE reads to CEGMA and ESTs. To assess the genome connectivity and assembly accuracy, we aligned 34,932 previously published mate-paired BAC-end sequences (BES) derived from Songpu mirror carp to the new assemblies[49,50].

We constructed high-density genetic linkage maps for HB and GM by genotyping F1 mapping families on the high-density 250 K SNP array following Affymetrix protocols. The double pseudo-test cross strategy was employed for linkage analysis using JoinMap 4.1 (https://www.kyazma.nl/). Recombination frequencies of markers on the same LG were converted into map distances (cM) through the maximum likelihood (ML) algorithm. The consensus map was then established using the MergeMap by integrating sex-specific maps through shared markers. All genetic linkage maps were drawn using MapChart 2.2[51]. Together with previously published high-density genetic linkage maps of YR, we have three high-density linkage maps available for scaffold integration.

To anchor scaffolds to each linkage map, we aligned the SNP-associated sequences of high-density genetic linkage maps to the assembled genomes using BLAST. Only SNP markers with a unique location were used for anchoring and orienting scaffolds. For those scaffolds that were in conflict with the genetic map, we performed manual checks using mate-paired reads.

**Gene prediction and functional annotation**. Gene prediction and functional annotation were performed through a combination of homology-based prediction, de novo prediction and transcriptome-based prediction methods. Protein sequences from *Ctenopharyngodon idella*, *Cynoglossus semilaevis*, *Denio rerio*, *Oryzias latipes*, *Tetraodon nigroviridis*, *Sinocyclocheilus graham*, *Sinocyclocheilus rhinocerous*, *Sinocyclocheilus anshuiensis*, *Mus musculus*, and *Homo sapiens* were aligned to the carp genome using TblastN (E-value < = 1e-5). The BLAST hits were conjoined by GeneWise (v2.4.1)[52] for accurate spliced alignments. For de novo prediction, three de novo prediction tools, Augustus (v2.7)[53], GlimmerHMM (v3.02)[54] and SNAP (version 2006-07-28)[55], were used to predict the genes in the repeat-masked genome sequences. The RNA-seq reads from multiple tissues were mapped onto the genome assembly using Tophat (v2.1.0)[56], and then Cufflinks (v2.1.1)[57] was used to assemble the transcripts into gene models. Gene predictions from the de novo approach, *homology-based* approach and RNA-Seq-based evidence were merged to form a comprehensive consensus gene set using the software EVM[58]. To achieve the functional annotation, the predicted protein sequences were aligned against public databases, including SwissProt, TrEMBLE and KEGG with BLASTP (E-value < = 1e-5). Additionally, protein motifs and domains were annotated by searching the InterPro and Gene Ontology (GO) databases using InterProScan (v4.8)[59].

**Repetitive element annotation**. Transposable elements in carp genomes were detected by combining homology-based and de novo predictions. For homology-based detection, RepeatMasker and RepeatProteinMask were used to screen the

carp genome for known transposable elements against the RepBase library (v20140131) (http://www.repeatmasker.org/). De novo transposable elements in the genome were identified by RepeatMasker based on a de novo repeat library constructed by RepeatModeller and LTR_FINDER (v1.0.5)[60]. Tandem repeats were detected using the program Tandem Repeats Finder (TRF, v4.07b)[61] with default parameters.

**RNA sequencing**. Twelve tissues (brain, muscle, liver, intestine, blood, head kidney, kidney, skin, gill, spleen, gonad and heart) were dissected and collected from six Yellow River carp. Total RNA was extracted from 12 tissues using TRIZOL (Invitrogen, Carlsbad, CA, USA). The RNA samples were then treated by DNase I. The integrity and size distribution were checked with Bioanalyzer 2100 (Agilent technologies, Santa Clara, CA, USA). The high-quality RNA samples were sequenced on Illumina HiSeq 2000 platforms with the manufacturer's instructions. A total of 72.2 Gb clean reads were generated for 12 tissues for expression analysis. Similarly, we also collected 75.6 Gb RNA reads of the same 12 tissues from grass carp *C. idella*, which has the complete reference genome available and serves as a diploid reference in this study. The RNA-seq data for biotic and abiotic stresses treatment were collected from SRA databases (Accession No. PRJNA314552 for CyHV-3 infection, Accession No. PRJNA315069 for *Aeromonas hydrophila* infection, and No. PRJNA512071 for hypoxia experiment).

**Phylogenetic analysis**. To explore the evolutionary relationship of *C. carpio* and its closely related tetraploid and diploid Cyprininae species, we constructed an ML phylogenetic tree based on *rag2* genes selected from a previous published dataset[12] or extracted from published Cyprinid genomes using MEGA7 software[62]. Four diploid Cyprininae species (*Poropuntius huangchuchieni*, *Hampala macrolepidota*, *Onychostoma barbatulum*, and *Cirrhinus molitorella*) representing three closely related diploid clades were selected for further investigation.

To confirm the homoeologous relationship of the two subgenomes in *C. carpio*, we built phylogenetic trees based on 2071 conserved homoeologous gene pairs among two tetraploids (*C. carpio* and *S. anshuiensis*) and their orthologues from three diploids (*D. rerio*, *C. idella* and *P. huangchuchieni*) using RAxML software[63] and integrated 2071 trees using programs MP-EST[64] and summary tree based on their topologies.

**Sample collection and genome sequencing of four diploid Cyprinid species**. Four diploid Cyprinid species that are closely related to allotetraploid common carp, *P. huangchuchieni*, *H. macrolepidota*, *O. barbatulum*, and *C. molitorella*, were collected for whole genome sequencing and draft assembly. *P. huangchuchieni* and *H. macrolepidota* were collected at Xishuangbanna in Yunnan Province, China, *O. barbatulum* was collected at Lishui in Zhejiang Province, China, and *Cirrhinus molitorella* was collected at Guangzhou, Guangdong Province, China. Genomic DNA was extracted from a fin clip or muscle using QIAGEN DNeasy Blood & Tissue Kit. The 350-bp whole-genome libraries were constructed for each sample according to the manufacturer's specifications (Illumina). The libraries were then sequenced on the Illumina HiSeq platform to generate raw sequences with a 150-bp read length. A total of 54.68 Gb (*Cirrhinus molititorella*), 57.35 Gb (*Poropuntius huangchuchieni*), 64.31 Gb (*Onychostoma barbatula*) and 71.78 Gb (*Hampala macrolepidota*) data were generated for the above four species and were subjected for primary genome assemblies using SOAP*denovo*. The genome sequences of the four newly sequenced diploid Cyprinids and two previously sequenced diploid Cyprinids (*Ctenopharyngodon idella* and *D. rerio*) were mapped to 25 pairs of homoeologous chromosomes of *C. carpio* genomes using BWA program. The loci with more than four mapped reads were considered effective. The genome coverages were calculated based on effectively covered genome regions, and then plotted with R package.

**Defining the conserved homoeologous gene pairs**. To determine the conserved homoeologous gene pairs in the tetraploid genome of *C. carpio*, we used annotated protein coding genes of the *C. idella* genome as a diploid reference. The OrthoMCL pipeline[65] was used to define gene families in the common ancestor. The all-against-all similarities were determined using blastp with an E-value cutoff of 1e-5. The orthologous triplets with 1:1:1 relationship in *C. idella* genome, subgenomes A and B of *C. carpio* were identified from two genomes. The orthologous gene pairs in the *C. carpio* genome were then defined as homoeologous genes that were derived from the latest WGD event.

**Homoeologous gene expression analysis**. Clean reads of RNA-seq were mapped onto the reference genome of *C. carpio* using Hisat2 (v2.0.4)[66]. Gene expressed level in terms of reads per kilobase of transcript per million mapped reads (RPKM) was estimated by HTseq[67] and custom Perl scripts. A gene was classified as 'expressed' if the FPKM value of at least one tissue was above 1.0, and values of genes were transformed to log2(FPKM + 1) values for consecutive analysis[13]. Genes were clustered using Pearson's correlation and Ward's method in the R function hclust, and visualized as heatmaps using the R function heatmap (ggplot2). Genes were scaled individually in the heatmaps.

Clusters with a significant number of shared homoeologous pairs were identified by simulation of 10,000 randomizations. A total of 8214 expressed

homoeologous genes were included in the analysis. A *C. carpio* gene was classified as conserved if the *P* value of Wilcoxon test to the *C. idella* orthologue was above 0.05 across the 12 common tissues, and diverged if the *P* value was below 0.05.

A homoeolog expression dominance analysis was performed for syntenic gene pairs. Differentially expressed genes pairs with greater than two-fold change were defined as dominant gene pairs. The dominant genes were the genes that were expressed relatively higher in dominant gene pairs, and the lower ones were the subordinate genes. The rest of the syntenic gene pairs that showed non-dominance were classified as neutral genes.

**Sequence evolution and divergence time analyses**. We identified each zebrafish-common carp orthologous sequence pairs and performed the Ka, Ks, and Ka/Ks test of the aligned sequence pairs. Briefly, orthologous protein sequences from zebrafish and common carp were aligned using ClustalW under default parameters. We estimated the number of synonymous and non-synonymous substitutions/sites using the codeml algorithms implemented in PAML package. We used the nucleotide substitution rate of $3.51 \times 10^{-9}$ per site per year from synonymous sites in coding regions as molecular clock[9]. Time of divergence was calculated using the formula $T = K/2r$, where K is the number of substitutions per base between sub-genomes and r is the rate of substitution. PercDivs (percentage of substitutions in the matching region compared to the consensus) of two subgenomes was calculated in RepeatMasker to assess TE divergence.

The non-overlapped segregation region indicates the time frame from diploid progenitor divergence (Ks-based estimation of 23 Mya) to progenitor diploid genomes merging as allotetraploid genome. The two time-points are corresponded to TE divergence rates of 13% and 7%, respectively. Therefore, the allotetraploidization time (genome merger time) is estimated as: 23 Mya× 7/13 = 12.4 Mya.

**Whole genome bisulfite sequencing (WGBS)**. Genomic DNA was extracted from muscle tissue of three common carp individuals. Genomic DNA was treated using a sodium bisulfite, which converts unmethylated cytosine to uracil, then thymine[68]. Whole-genome bisulfite sequencing (WGBS) was performed using an Illumina HiSeq2500 sequencer with 150 bp paired-end sequences (Illumina, San Diego, CA, USA).

WGBS reads were mapped to genome sequences of *C. carpio*. Bismark with parameters (-score_min L,0, -0.2 -X 700 -no-mixed -no-discordant -dovetail) was used for read mapping. Only reads mapped to the unique sites were retained and used for further analysis. The reads mapped to the same site were collapsed into a single consensus read to reduce clonal bias. To avoid the base bias among biological replicates, only the conserved cytosines were used for the analysis. All conserved cytosines were called when the same bases were covered by at least five reads in all three replicates and used for further analysis.

**Defining body-methylated genes**. Only cytosines covered by at least five reads in the three replicates were selected for downstream analysis. The methylation level of a cytosine site was calculated as $C/(C + T)$[69]. C indicates the number of reads with cytosine for this site. T indicates the number of reads with thymine for this site. The mCG sites are recognized by using a binomial distribution test. The level of DNA methylation was calculated for each protein-coding region according to the previous published article[43]. Briefly, the levels of DNA methylation in CG contexts are calculated as follow equation:

$$PCG = \sum_{i=mcg}^{ncg} \binom{ncg}{i} pcg^i (1 - pcg)^{ncg-i} \qquad (1)$$

where *PCG* is a proxy *P* value of DNA methylation level, *pcg* is the proportion of methylated cytosine residues at CG sites across the whole genome, *ncg* and *mcg* are the number of cytosine residues at CG sites with ≥5 coverage and the number of methylated cytosine residues at CG sites in a gene, respectively. Assuming a binomial probability distribution, the one-tailed *P* value for the departure of CG methylation level from genome average was then calculated. To test the relationship between methylation and sequence evolution in genic regions, we identified CG body-methylated genes (*PCG* <0.05) in the 8291 homoeologous gene pairs. The total cytosine residues at CG sites across the genome is 42,911,370 and three biological replicates share a total of 21,275,948 methylated CG sites. Then pcg is calculated as 49.58%. Finally, 2393 from the 8291 homoeologous genes pairs are selected to investigate the relationship between CG methylation changes and sequence neutral substitution changes in genic regions. Differentially methylated CGs (DmCGs) in 2393 homoeologous gene pairs are counted when the conserved CG sites are methylated in one homoeolog but unmethylated in the other in all three biological replicates. The methylation change rate is calculated based on the number of DmCGs among all conserved CG sites.

**Reporting summary**. Further information on research design is available in the Nature Research Reporting Summary linked to this article.

## Data availability

All genomic sequence datasets of three distinct *C. carpio* strains can be found on NCBI (https://www.ncbi.nlm.nih.gov/bioproject/PRJNA510861/) and BIGD Genome Warehouse (https://bigd.big.ac.cn/bioproject/browse/PRJCA001408). Genome resources of four closely- related diploid Cyprinid genomes are also available on NCBI (https://www.ncbi.nlm.nih.gov/bioproject/PRJNA511029); (https://www.ncbi.nlm.nih.gov/bioproject/PRJNA511030); (https://www.ncbi.nlm.nih.gov/bioproject/PRJNA511031); (https://www.ncbi.nlm.nih.gov/bioproject/PRJNA511032).

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

## Acknowledgements

We acknowledge grant support from the National Natural Science Foundation of China (grants 31422057 & 31872561 to P.X. and 31502151 to J.X.), the National High-Technology Research and Development Program of China (grant 2011AA100401 to P.X.), the Fundamental Research Funds for the Central Universities, Xiamen University (grants 20720180123 & 20720160110 to P.X.), Central Public-interest Scientific Institution Basal Research Fund, CAFS (No. 2016HY-JC0301 to J.X. and 2017B003 to H.Z.) and the National Key Research and Development Program (2018YFD0900102 to J.X.).

## Author contributions

P.X. conceived and designed the research. P.X. and J.X. coordinated the project. G.L. and W.J. developed the sequencing strategy, performed next-generation sequencing, assembly and bioinformatics. J.X., L.C., Z. Zhao, and C.D. performed RNA-Seq collection and DNA methylation sequencing. P.X., J.X., L.C., Z. Zhou, Y.J., Z. Zhao, B.C., F.P., J.F., J.L., J.C., H.W., and C.D. collected and prepared the samples. J.X., L.C., W.P., and Z.J. performed the genotyping and genetic linkage mapping. Z. Zhou, W.P., and L.C. performed phylogenetic analysis. J.X., B.C., H.W., Y.W., Y.S., and H.Z. performed wet-lab molecular experiments. P.X., L.C., G.L., and J.X. wrote and revised the manuscript and the supplementary information. X.S. participated in discussions and provided valuable advice.

## Competing interests

The authors declare no competing interests.
