## [Peer Review File · Nature Communications]

Reviewers' Comments:

Reviewer #1:

Remarks to the Author:

Genome duplication events are seen as major drivers of increasing genome complexity in species ranging across yeast, plants and animals. But most of these events, and the associated processes, have been 'camouflaged' by substantial gene losses and re-diploidisation back to the 'sustainable' gene composition. However, a small group of Cyprinid species, comprising the common carp, the Crucian carp and the goldfish, possess a tetraploid karyotype of 100 chromosomes due to a single allo-tetraploidisation event 10-12MYA. Xiaowen Sun and Peng Xu have led the use of DNA sequencing to understand this latter situation. They published the common carp genome sequence in 2014, which beautifully demonstrated paired sets of chromosomes, each of which had retained gene synteny relative to the distant zebrafish genome, with surprisingly little evidence of gene loss.

This new manuscript now clearly differentiates the sets of homeologous chromosomes into two distinctive sub-genomes. The authors demonstrate a close sequence similarity of one set of chromosomes to an extant, diploid species, which is inferred to be a descendant of one of the two progenitor lineages that contributed to the allotetraploidisation event. They demonstrate distinctive substitution dynamics of one sub-genome relative to the other, and detect greater levels of gene expression for one sub-genome relative to the other. These characteristics offer some insight into the manner of evolution of this very recent tetraploidy, which is distinct from that shown in the earlier salmonid duplication event (~60MYA) which since has led to re-diploidisation.

In my view, this is a very significant new paper, first because it gives unprecedented detail relating to the outcome of a comparatively recent evolutionary event in a vertebrate animal; second, because it takes advantage of accessing a good number of candidate progenitor species to define likely relationships within each sub-genome; and third, because it promotes the wider exploration of separately evolved functional roles of genes in each of the sub-genomes, especially in serving an expanded range of adaptive features. This paper will interest a wide variety of scientists with an interest in evolutionary mechanisms.

This manuscript presents a very substantial body of new information, having sequenced several members of the common carp species, but also of several closely related diploid species. Key evidence for differentiation of the two sub-genomes comes from divergent sequence relationships across three diploid progenitor species within the species-rich Cyprininae. One of these is especially close in DNA sequence to one of the proposed common carp sub-genomes. This and other evidence is entirely consistent with a allo-tetraploidy event.

An unresolved issue in identifying the beneficial outcomes of the duplication event, is the extent to which the contributory sub-genomes had diverged before or after the duplication event, and whether duplication enabled increased rates of evolutionary sequence divergence. The graded relationship of sub-genome B with several diploid carp lines indicates that at least some sequence features predated the duplication event, in which case the phenotypic benefits of duplication might arise from divergence of gene functions endowing greater levels of environmental adaptation. One unaddressed question is, has the functional divergence of genes/proteins subsequent to duplication extended the differences between species prior to duplication. In other words, does the combination of sub-genomes allow additive functional benefits to evolve that improves functional competency.

A related issue is just how robust is their assignment of sub-genomes given the likelihood of increasing numbers of related Cyprinid species being fully sequenced. At present only one of these sub-genomes has been linked extant species, and lack of corroboration from the alternate progenitor

puts the onus on the authors to exclude potential pitfalls in their chosen interpretation. Thus, co-location of two nearly identical sub-genomes from a tetraploidisation event is possibly followed by different rates of sequence/functional divergence in the two sub-genomes. This might well make it more likely that the less diverged sub-genome in the common carp shows a closer phylogenetic relationship with genes in the progenitor B lineage, even though it may actually stem from progenitor A. Such a problematic assignment could potentially confound the subsequent analysis of expression bias. Perhaps this kind of complication can be considered.

Earlier work by others has demonstrated the extraordinary functional divergence of duplicated genes in carp. The assumption is that co-expression of diverged, sub-functionalised proteins is a key contributor to the unusually high levels of stress tolerance exhibited by the common carp to thermal extremes, hypoxia and anoxia. But what is not clear from this manuscript is whether this divergence occurred before or after tetraploidisation. In the former case, genes of the ancestral species had diverged functionally prior to tetraploidisation, whilst in the latter case, divergence occurred after tetraploidisation to allow co-expression of functionally differentiated proteins.

In several places, the text specifically links the duplication event to an uplift in the Mongolian plateau. To my understanding this is speculative, but not necessarily unlikely, since timing of both geological and duplication events will inevitably be approximate.

Specific questions –

1. Figure 1 comprises a phylogeny of species relevant to this paper. Fig 2a comprises a similar if not identical display of phylogenetic relationships of species. Is there a need to duplicate this information?
2. Figure 2. Use of the word 'pentagram' in the legend for Fig 2a is confusing, at least to me, because I don't see 5 elements to what is a conventional display of evolutionary relationships. Also have the authors made the sequence data for these species openly available to the research community. Regarding Fig 2b, I am confused as to why values for A chromosomes are so much lower than for sub-genome B. Claims of a conserved synteny seem inconsistent with the huge difference in coverage. For Figure 2e, the associated text indicates TE sequence divergence between the sub-genomes, yet the figure displays an amazing degree of overlap.
3. Figure 3a displays the beautiful gene-gene relationship between genes present in the two proposed sub-genomes. This supersedes the same figure in their 2014 paper, having resolved some remaining issues of gene allocation between sub-genomes in that earlier figure. Fig 3d, displays a complex image of heatmaps for (presumably) three groups of genes, across three (sub-)genomes. Will this data be made available in supplementary material and, preferably, online? Second, exactly what does the data represent and how have expression values been normalised? The adjacent colour-coded panel has no indication of this meaning. In Fig 3c is the colour coding of increased and decreased expression linked to the same colour coding in (b)? If so then it implies that all sub-genome B genes were downregulated and the exact reverse for sub-genome A.
4. Figure 4 has mislabelled some of the sub-figures. Thus, the heatmaps are indicated in the fig as (c) but listed in the legend as (b). Also some figures lack labelling of the axes. A good example is Fig 4a which lacks a formal label for the abscissa. Fig 4d includes TSS on the ordinate, but this should be defined in the legend. Fig 4e lacks a label for the ordinate axis. Lack of this information can easily confused some readers and force them to search through the rest of the document.

Reviewer #2:

Remarks to the Author:

The manuscript describes the sequencing of the tetraploid common carp *C. carpio* genome. Three

different strains have been sequenced to high completion, representing a substantial incremental progress over the existing but fragmented version of the common carp assembly already published by the same authors in 2014. More importantly, the authors produced four draft assemblies of diploid species thought to be sister groups of *C. carpio* that diverged before the tetraploidisation. Remarkably, the authors find that the diploid species align only to one half of the chromosome pairs of *C. carpio*, thus suggesting that this half corresponds to one distinct subgenome originating from the allotetraploidisation. The authors then provide a date estimate of the tetraploidisation and discuss the possible geo-climatic context of the event. They then provide genome wide expression and methylation data in support of the surprising finding that one subgenome may be dominant (in terms of gene retention and gene expression levels) over the other one. Finally, they provide a somewhat disconnected set of results related to the genetic determinants of the red colour of some *C. carpio* strains.

Overall I find the study very interesting and well structured. The new resource is welcome, and the dominant genome model is very exciting because it had always been thought (to my knowledge) that this was a specific feature of plant genomes. For example the salmonid specific 4R did not reveal any dominance pattern, nor did the teleost 3R leave any signature of such a phenomenon. I find the data supporting this model convincing on the whole, especially with the promoter methylation. On the other hand, specifically regarding the excess of gene retention of subgenome B versus A, I think this is not sufficiently robust to remain as evidence in the manuscript (see below) unless it is statistically supported. Although I am generally enthusiastic about the manuscript, I find that some parts need substantial work. The section on the geo-climatic interpretation (coincidence of the tetraploidisation with the uplift of the Qinghai-Tibetan Plateau) seems to me highly speculative. It would be better to tone down the way this association is presented and leave it to the discussion. Figure 4 needs a lot more clarity. Regarding the genetic determinants of the red colour, I would suggest replacing it in a study of its own. It requires more extensive demonstrations to be really convincing and a dedicated article would be more appropriate for this.

Specific points:

1. P4L106-107 : references are inadequate. Ref 1-4 describe mostly the Carp 4R but are used to describe the vertebrate 1R-2R and the teleost 3R. Similarly Ref 5 is used to describe the teleost 3R but the correct one should probably be Jaillon et al. Nature 2004.
2. P4L116: The date of the salmonid duplication should be revised and ought to cite Macqueen and Johnston 2014. Proc. R. Soc. B. 281: 20132881 and Berthelot et al. Nat Commun. 2014.
3. P4L130: I do not understand why the authors say "polyploid genomes involve multiple rounds of WGD and segmental duplications that harbour much more complex structures and gene contents than..." What is the evidence in support of this statement?
4. P32-L894: Data availability: Currently the NCBI accessions given in the manuscript correspond to just the raw reads, and I noted that the current genome assemblies available from NCBI are for unplaced scaffolds. This manuscript relies heavily on the fact that the 3 genome assemblies are well anchored to chromosomes thanks to the genetic maps. Therefore the authors should provide the NCBI accession IDs of the assembled genome sequences (with chromosome anchoring).
5. P8L228: Why choose rag2? It is not clear to me why this gene particularly. How did the authors build the phylogeny? Figure 2a: branch lengths would be very useful to help interpret the phylogeny. Same for Supp. figure 7.
6. P9L271: how can the authors firmly state that the donor diploid species of subgenome B originated in South West China? Are extant Barbinae only found in this region? There are no fossils in other regions? This section of the manuscript appears very speculative and should be exclusively mentioned as hypothetical, in the discussion.
7. Page 10: I do not understand Figure 2e and the related text. The rationale for the interpretation of

the "bubble" should be better explained, as this is the main evidence for the dating of the allotetraploidy. Also the computation of the TE substitution rates (line 298) leading to the date of 12.4 Mya should be better explained, with a specific section in Methods.

8. P10L301-309: Please tone down or remove the speculative geological interpretation. Diploid species survived these environmental events just fine.

9. P11L336: Could the authors justify here their choice of *C. idella* as outgroup of the Carp 4R? The reason only comes much later page 14.

10. P11L31 and after: The author's early conclusion of dominance based on gene retention rates is not well supported. Based on gene loss, the relative difference is that subgenome A lost 2.16% of duplicated copies and subgenome B lost 2.88% (Supplementary figure 10). Unless the authors can back this up with some meaningful statistics, this should be considered as completely expected under a random process and therefore the authors should remove any mention of dominance in the manuscript based on this evidence alone.

11. P12L360: Could the authors verify that BAC-end sequences span the breakpoints in the two subgenomes and that they can therefore exclude any assembly errors, that could be an alternative explanation for the rearrangements?

12. P13L379: the authors indicate mean $Ka/Ks = 0.20$ but according to Supp Table 26 this should be "median Ka/Ks ". Is this still based on the 10,274 pairs and triplets of genes with *C. idella* as outgroup? Please clarify.

13. P13: The authors do not know precisely which clade the donor of subgenome A originates from. It may be evolutionarily more distant to *C. idella* than the donor of subgenome B. The authors should discuss the implications of these differences for evolutionary rate estimations. If population sizes, generation times, etc imposed a different molecular clock on the donor of subgenome A prior to the allotetraploidisation, then surely this could conceivably generate differences in Ka/Ks ratio (even if theoretically it should not) between the two subgenomes when measured today. It would be good if the authors could think of a control to rule out that this is the reason behind their observation. For example, comparing TE evolutionary rates might be useful in this regard.

14. P14L414. I only see 6 clusters in Supplementary Figure 13, not 8.

15. P14L415: I do not understand what the authors mean by "assigned to different co-expression clusters" nor later with "spatial partitioning".

16. Figure 3d: Please indicate how many genes are comprised in the 3 clusters. How do these clusters relate to those in Supp Figure 13?

17. P15L441 and Figure 4c. The legend of Figure 4 says that the two highly divergent clusters were defined on the basis of one of the two homeologs being "completely silenced in all 12 tissues." When looking at Figure 4c this appears not to be the case, some genes in the upper right panel (extinguished in B) are actually expressed, some very highly so. Please explain this experiment more rigorously.

18. Figure 4: please use different colour scheme than red/blue for everything throughout the figure. Blue means B in 4e but A in 4b/d and methylation in 4a and low expression in 4c. It does not help with clarity.

19. P15L445: The authors must mean Figure 4e

20. P18L542: I do not understand Figure 4a. How can the authors conclude that methylation change was "faster"? The authors seem to compare the absolute ratio of synonymous changes (Ks) with a percentage of methylation change. This does not make immediate sense to me because while Ks is indeed an approximation of evolutionary time, the % methylation is definitely not (there is no indication that it varies at a constant rate with time). Could the authors clarify this? Also, the legend mentions arrows for peak values, but I cannot see them.

21. P19L556: what test was performed to compute this p value?

22. P19L556-560: repetition of the previous sentence.

23. P20L586: please annotate better Figure 4f: what is the horizontal arrow with vertical bars, what are the black bars in the eif3ia/b diagrams? The text mentions two additional genes that are absent

for the figure.

24. P22 and Figure 5a and 5c: the rationale for selecting these two genes based on the data is not clear.

25. P22: what is the expression of *bco2a-1* in control (i.e. grey) *C. carpio* strains? This data should be provided to conclude that this gene is causative.

26. I believe that homoelogenous/homoelogs is misspelled throughout the manuscript (it should be homeologous).

Point-to-Point Response to Reviewers' Comments:

Reviewer #1 (Remarks to the Author):

Genome duplication events are seen as major drivers of increasing genome complexity in species ranging across yeast, plants and animals. But most of these events, and the associated processes, have been 'camouflaged' by substantial gene losses and re-diploidisation back to the 'sustainable' gene composition. However, a small group of Cyprinid species, comprising the common carp, the Crucian carp and the goldfish, possess a tetraploid karyotype of 100 chromosomes due to a single allo-tetraploidisation event 10-12MYA. Xiaowen Sun and Peng Xu have led the use of DNA sequencing to understand this latter situation. They published the common carp genome sequence in 2014, which beautifully demonstrated paired sets of chromosomes, each of which had retained gene synteny relative to the distant zebrafish genome, with surprisingly little evidence of gene loss.

This new manuscript now clearly differentiates the sets of homeologous chromosomes into two distinctive sub-genomes. The authors demonstrate a close sequence similarity of one set of chromosomes to an extant, diploid species, which is inferred to be a descendant of one of the two progenitor lineages that contributed to the allotetraploidisation event. They demonstrate distinctive substitution dynamics of one sub-genome relative to the other, and detect greater levels of gene expression for one sub-genome relative to the other. These characteristics offer some insight into the manner of evolution of this very recent tetraploidy, which is distinct from that shown in the earlier salmonid duplication event (~60MYA) which since has led to re-diploidisation.

In my view, this is a very significant new paper, first because it gives unprecedented detail relating to the outcome of a comparatively recent evolutionary event in a vertebrate animal; second, because it takes advantage of accessing a good number of candidate progenitor species to define likely relationships within each sub-genome; and third, because it promotes the wider exploration of separately evolved functional roles of genes in each of the sub-genomes, especially in serving an expanded range of adaptive features. This paper will interest a wide variety of scientists with an interest in evolutionary mechanisms.

This manuscript presents a very substantial body of new information, having sequenced several members of the common carp species, but also of several closely related diploid species. Key evidence for differentiation of the two sub-genomes comes from divergent sequence relationships across three diploid progenitor species within the species-rich Cyprininae. One of these is especially close in DNA sequence to one of the proposed common carp sub-genomes. This and other evidence is entirely consistent with a allo-tetraploidy event.

An unresolved issue in identifying the beneficial outcomes of the duplication event, is the extent to which the contributory sub-genomes had diverged before or after the duplication event, and whether duplication enabled increased rates of evolutionary sequence divergence. The graded relationship of sub-genome B with several diploid carp lines indicates that at least some sequence features predated the duplication event, in which case the phenotypic benefits of duplication might arise from divergence of gene functions endowing greater levels of environmental adaptation. One unaddressed question is, has the functional divergence of genes/proteins subsequent to duplication extended the differences

between species prior to duplication. In other words, does the combination of sub-genomes allow additive functional benefits to evolve that improves functional competency.

A related issue is just how robust is their assignment of sub-genomes given the likelihood of increasing numbers of related Cyprinid species being fully sequenced. At present only one of these sub-genomes has been linked extant species, and lack of corroboration from the alternate progenitor puts the onus on the authors to exclude potential pitfalls in their chosen interpretation. Thus, co-location of two nearly identical sub-genomes from a tetraploidisation event is possibly followed by different rates of sequence/functional divergence in the two sub-genomes. This might well make it more likely that the less diverged sub-genome in the common carp shows a closer phylogenetic relationship with genes in the progenitor B lineage, even though it may actually stem from progenitor A. Such a problematic assignment could potentially confound the subsequent analysis of expression bias. Perhaps this kind of complication can be considered.

Earlier work by others has demonstrated the extraordinary functional divergence of duplicated genes in carp. The assumption is that co-expression of diverged, sub-functionalised proteins is a key contributor to the unusually high levels of stress tolerance exhibited by the common carp to thermal extremes, hypoxia and anoxia. But what is not clear from this manuscript is whether this divergence occurred before or after tetraploidisation. In the former case, genes of the ancestral species had diverged functionally prior to tetraploidisation, whilst in the latter case, divergence occurred after tetraploidisation to allow co-expression of functionally differentiated proteins.

In several places, the text specifically links the duplication event to an uplift in the Mongolian plateau. To my understanding this is speculative, but not necessarily unlikely, since timing of both geological and duplication events will inevitably be approximate.

Our response: We very appreciate for reviewer's positive comments and valuable suggestion to our manuscript. We absolutely agree with reviewer's comments and concerns on homoeologous gene expression divergence and subfunctionalisation. To solve this issue, it is critical importance to identify the closely related diploid extant progenitors and develop the high-quality reference genomes as outgroup. With more and more related Cyprinid genomes being fully sequenced, we believe that we will have more robust evidences for homoeologous chromosome assignment and gene divergence. On the other hand, the increasing transcriptome data for gene expression analysis of both homoeologous gene pairs of common carp and the orthologous genes of its diploid progenitors, under regular and high level of environmental stresses, would help us clarify whether the gene function divergence occurred prior to or post tetraploidisation. We recently submitted a research proposal that aim to develop high-quality reference genomes of the closely related diploid species including *Poropuntius huangchuchieni* and *Onychostoma barbatulum*, and to collect transcriptome and DNA methylation data during thermal stresses. It will provide us fundamental genome information and enable further exploration on genomic basis of evolutionary advantages and beneficial outcomes of the duplication event in allotetraploid carp. Regarding to the links between geological event and genome duplication of common carp, Reviewer #2 has similar comment and concerns. We therefore have removed the section about geological interpretation to prevent potential misleading.

Specific questions –

1. Figure 1 comprises a phylogeny of species relevant to this paper. Fig 2a comprises a similar if not identical display of phylogenetic relationships of species. Is there a need to duplicate this information?

Our response: We used Figure 1 to give a brief introduction of teleost specific 3R WGD, as well as the latest 4R duplication in teleost including Ss4R and Cs4R. The phylogenetic topologies and divergence ages are taken from the TimeTree database and previous published papers. So, Figure 1 is our basic knowledge before we started the project. Figure 2a was built based on RAG2 genes that collected from common carp and its closely related diploid and tetraploid Cyprinid species. It was the key step to discriminate two subgenomes of common carp. Figure 2a indicated that one of the subgenome (subgenome B) is very close to one of the diploid lineages. We therefore selected several diploid species from this lineage for whole genome sequencing, and further performed comparative genomic analysis and confirmed our hypothesis about diploid progenitors.

2. Figure 2. Use of the word 'pentagram' in the legend for Fig 2a is confusing, at least to me, because I don't see 5 elements to what is a conventional display of evolutionary relationships. Also have the authors made the sequence data for these species openly available to the research community. Regarding Fig 2b, I am confused as to why values for A chromosomes are so much lower than for sub-genome B. Claims of a conserved synteny seem inconsistent with the huge difference in coverage. For Figure 2e, the associated text indicates TE sequence divergence between the sub-genomes, yet the figure displays an amazing degree of overlap.

Our response: Thanks for the reminder. We missed the pentagram labels during multiple rounds of modification of Figure 2. Now we have revised Figure 2a as Reviewer #2's suggestion, and added pentagrams to mark four diploid species that we selected to sequence. We have all genome sequence data of these sequenced diploids uploaded to public available database (NCBI accession numbers: PRJNA511029-32).

Regarding to Figure 2b: we mapped short genome reads of closely related diploid species, *Poropuntius huangchuchieni*. We used mapping coverage on target chromosomes of common carp to evaluate the chromosome-wide similarity. While the high conserved synteny (as shown in Figure 3a) was build based the gene positions of homoeologous gene pairs, and the homoeologous gene pairing were based on conserved gene coding sequences. The gene coding sequences are usually significantly more conserved than random genome sequences. Therefore, Figure 2b illustrated significant differences on genome mapping coverage.

Regarding to Figure 2e: The dating of allotetraploidization was calculated based on two approaches. 1) The synonymous substitution rates (Ks) of homoeologous genes was used to determine the divergence time of the two progenitors (two progenitors diverged from their common ancestor around 23 Mya). 2) The distribution of TE sequence substitution rate reflects TE divergence of two subgenomes. Despite of high degree of overlap, we still identified TE content segregation between subgenomes A and B (non-overlapped "bubble" region) as shown in shaded region in Figure 2e. The non-overlapped segregation region indicates the time frame from diploid progenitor divergence (Ks-based estimation of 23 Mya) to progenitor diploid genomes merging as allotetraploid genome. The two time-points are corresponded to TE divergence rates of 13% and 7%, respectively. Therefore, the allotetraploidization time (genome merger time) is estimated as: $23 \text{ Mya} * 7/13 = 12.4 \text{ Mya}$.

3. Figure 3a displays the beautiful gene-gene relationship between genes present in the two proposed sub-genomes. This supersedes the same figure in their 2014 paper, having resolved some remaining issues of gene allocation between sub-genomes in that earlier figure. Fig 3d, displays a complex image

of heatmaps for (presumably) three groups of genes, across three (sub-)genomes. Will this data be made available in supplementary material and, preferably, online?

Our response: Thanks for the reminder. We have listed the homoeologous gene pairs of Figure 3a and gene expression values related to Figure 3d as supplementary tables (Supplementary Table 31 and 64). The supplementary files will be available online once published.

Second, exactly what does the data represent and how have expression values been normalised? The adjacent colour-coded panel has no indication of this meaning. In Fig 3c is the colour coding of increased and decreased expression linked to the same colour coding in (b)? If so then it implies that all sub-genome B genes were downregulated and the exact reverse for sub-genome A.

Our response: We normalized the gene expression value using $\log_2(\text{FPKM})$. We used $\text{Log}_2(\text{FPKM}_A/\text{FPKM}_B)$ to indicate the degree of expression difference of homoeologous gene pairs in Figure 3c. We have added the details into Method section for clarifying. We have added x-axis label as "Log2(FPKM_A/FPKM_B)" in Figure 3c, and modified x-axis label in Figure 3e accordingly. We have changed colours in Figure 3b to prevent potential confusing with colours in Figure 3c. Figure legends have been revised accordingly.

4. Figure 4 has mislabelled some of the sub-figures. Thus, the heatmaps are indicated in the fig as (c) but listed in the legend as (b). Also some figures lack labelling of the axes. A good example is Fig 4a which lacks a formal label for the abscissa. Fig 4d includes TSS on the ordinate, but this should be defined in the legend. Fig 4e lacks a label for the ordinate axis. Lack of this information can easily confused some readers and force them to search through the rest of the document.

Our response: We have modified Figure 4 and corrected all mistakes and missing labels. Thanks.

Reviewer #2 (Remarks to the Author):

The manuscript describes the sequencing of the tetraploid common carp *C. carpio* genome. Three different strains have been sequenced to high completion, representing a substantial incremental progress over the existing but fragmented version of the common carp assembly already published by the same authors in 2014. More importantly, the authors produced four draft assemblies of diploid species thought to be sister groups of *C. carpio* that diverged before the tetraploidisation. Remarkably, the authors find that the diploid species align only to one half of the chromosome pairs of *C. carpio*, thus suggesting that this half corresponds to one distinct subgenome originating from the allotetraploidisation. The authors then provide a date estimate of the tetraploidisation and discuss the possible geo-climatic context of the event. They then provide genome wide expression and methylation data in support of the surprising finding that one subgenome may be dominant (in terms of gene retention and gene expression levels) over the other one. Finally, they provide a somewhat disconnected set of results related to the genetic determinants of the red colour of some *C. carpio* strains.

Overall I find the study very interesting and well structured. The new resource is welcome, and the dominant genome model is very exciting because it had always been thought (to my knowledge) that this was a specific feature of plant genomes. For example the salmonid specific 4R did not reveal any

dominance pattern, nor did the teleost 3R leave any signature of such a phenomenon. I find the data supporting this model convincing on the whole, especially with the promoter methylation. On the other hand, specifically regarding the excess of gene retention of subgenome B versus A, I think this is not sufficiently robust to remain as evidence in the manuscript (see below) unless it is statistically supported. Although I am generally enthusiastic about the manuscript, I find that some parts need substantial work. The section on the geo-climatic interpretation (coincidence of the tetraploidisation with the uplift of the Qinghai-Tibetan Plateau) seems to me highly speculative. It would be better to tone down the way this association is presented and leave it to the discussion. Figure 4 needs a lot more clarity. Regarding the genetic determinants of the red colour, I would suggest replacing it in a study of its own. It requires more extensive demonstrations to be really convincing and a dedicated article would be more appropriate for this.

Our response:

Thank reviewer for the positive comments and suggestion. Regarding the genetic determinants of the red colour in Hebao red carp, we agree with the reviewers that we should provide more convincing evidences. We have removed this section and Figure 5 from this submission. We will perform gene function verification experiments (i.e. using CRISPR techniques) of both *bco2* and *oca2* genes in common carp. We will submit another article once we collect more convincing evidences to demonstrate the genetic basis of red colour of Hebao red carp. We agree with reviewer's comment on gene retention and gene losses of two subgenome. Slight difference on gene losses between two subgenomes does not support asymmetric gene losses and subgenome dominance of gene content. We have modified the paragraph and removed the related sentences.

Regarding to geo-climatic interpretation, both reviewers point out that the links between geological event and genome duplication of common carp are speculative and we agree with them. We have removed the section about geological interpretation to prevent potential misleading.

Specific points:

1. P4L106-107: references are inadequate. Ref 1-4 describe mostly the Carp 4R but are used to describe the vertebrate 1R-2R and the teleost 3R. Similarly Ref 5 is used to describe the teleost 3R but the correct one should probably be Jaillon et al. Nature 2004.

Our response: The references have been replaced, and the new references have been cited. Thanks.

2. P4L116: The date of the salmonid duplication should be revised and ought to cite Macqueen and Johnston 2014. Proc. R. Soc. B. 281: 20132881 and Berthelot et al. Nat Commun. 2014.

Our response: The two new references have been added as suggested. Thanks.

3. P4L130: I do not understand why the authors say "polyploid genomes involve multiple rounds of WGD and segmental duplications that harbour much more complex structures and gene contents than..." What is the evidence in support of this statement?

Our response: We have re-phrased the sentence and removed the inaccurate description.

4. P32-L894: Data availability: Currently the NCBI accessions given in the manuscript correspond to just the raw reads, and I noted that the current genome assemblies available from NCBI are for unplaced scaffolds. This manuscript relies heavily on the fact that the 3 genome assemblies are well anchored to chromosomes thanks to the genetic maps. Therefore the authors should provide the NCBI accession IDs of the assembled genome sequences (with chromosome anchoring).

Our response: We have already uploaded three genome assemblies with chromosome anchoring information to NCBI database under Bioproject PRJNA510861. The NCBI submissions/updates, however, take more time than we expected. Currently, we are still waiting for the accession numbers of the three updated genome assemblies. The NCBI staff replied us that they may not complete the process very soon, but the updated genome assemblies will still be listed under the same Bioproject # PRJNA510861.

Considering the approaching revision due date, we submitted three chromosome-anchored genome assemblies to the alternative public genome databases of BIGD. Now three genome assemblies have been archived in Genome Warehouse database (<http://bigd.big.ac.cn/gwh/>) under BioProject # PRJCA001408 and already released for public access. The accession numbers of three genomes of Yellow River carp, German mirror carp and Hebao red carp are GWAATB00000000, GWAATC00000000, GWAATD00000000, respectively. We have included both Bioproject IDs of NCBI and BIGD in the revised manuscript for review and public access.

5. P8L228: Why choose rag2? It is not clear to me why this gene particularly. How did the authors build the phylogeny? Figure 2a: branch lengths would be very useful to help interpret the phylogeny. Same for Supp. figure 7.

Our response: The nuclear gene rag2 had been previously used in many phylogenetic analysis in Cyprinidae. It has been reported rag2 gene only appears as conserved single copy in diploid such as zebrafish. Wang and his colleagues recently published an article about polyploid origins in Cyprininae (Wang et al, 2016). In this study, they collected rag2 gene sequences from 123 species of the subfamily Cyprininae that distributed in 46 genera and 20 non-Cyprininae cyprinid species, and incorporated information of ploidy levels of these species. The article not only inspired us about allotetraploid evolution and origin studies on common carp, but also provide us rag2 sequences of closely related species. Our analysis further confirmed that rag2 gene presents only one copy in diploid cyprinids but two copies in tetraploid cyprinids.

We build the phylogeny in Figure 2a and Supplementary Figure 5 based on the rag2 sequences collected from Wang's article and newly developed genome data (mainly extracted from genome data of allotetraploid species including *C. carpio* and *Sinocyclocheilus*), to demonstrate the phylogenetic topology of allotetraploid *C. carpio* and its closely related diploids. We have added more details about the phylogenetic analysis into Methods for clarifying this. As suggested, we have built the new phylogenetic tree with branch length and replaced previous Figure 2a. Supplementary Figure 7 displays the integrated topologies based on 2,071 phylogenetic trees based on 2,071 genes using Bin-MPEST software. The branch length had been added as suggested.

Reference:

Wang X Z, Gan X N, Li J B, et al. Cyprininae phylogeny revealed independent origins of the Tibetan Plateau endemic polyploid cyprinids and their diversifications related to the Neogene uplift of the plateau. *Sci China Life Sci*, 2016, 59: 1149–1165, doi: 10.1007/s11427-016-0007-7

6. P9L271: how can the authors firmly state that the donor diploid species of subgenome B originated in South West China? Are extant Barbinae only found in this region? There are no fossils in other regions? This section of the manuscript appears very speculative and should be exclusively mentioned as hypothetical, in the discussion.

Our response: Sorry for the inaccurate statement. The extant Barbinae species have wide distribution in the world, not only in Asia but also in Europe and Africa. We have revised the section as suggestion and removed inaccurate statement.

7. Page 10: I do not understand Figure 2e and the related text. The rationale for the interpretation of the “bubble” should be better explained, as this is the main evidence for the dating of the allotetraploidy. Also the computation of the TE substitution rates (line 298) leading to the date of 12.4 Mya should be better explained, with a specific section in Methods.

Our response: The dating of allotetraploidisation was calculated based on two approaches. 1) The synonymous substitution rates (K_s) of homoeologous genes was used to determine the divergence time of the two progenitors (two progenitors diverged from their common ancestor around 23 Mya). 2) The distribution of TE sequence substitution rate reflects TE divergence of two subgenomes. Despite of high degree of overlap, we still identified TE content segregation between subgenomes A and B (non-overlapped "bubble" region) as shown in shaded region in Figure 2e. The non-overlapped segregation region indicates the time frame from diploid progenitor divergence (K_s -based estimation of 23 Mya) to progenitor diploid genomes merging as allotetraploid genome. The two time-points are corresponded to TE divergence rates of 13% and 7%, respectively. Therefore, the allotetraploidisation time (genome merger time) is estimated as: $23 \text{ Mya} * 7/13 = 12.4 \text{ Mya}$. We have added specific section in Methods as suggested.

8. P10L301-309: Please tone down or remove the speculative geological interpretation. Diploid species survived these environmental events just fine.

Our response: We have removed the section about geological interpretation according to the suggestion of both reviewers.

9. P11L336: Could the authors justify here their choice of *C. idella* as outgroup of the Carp 4R? The reason only comes much later page 14.

Our response: Thank you for the suggestion. We have added a brief introduction about choosing *C. idella* genome as diploid outgroup for common carp in the context, and revised the later paragraph accordingly.

10. P11L31 and after: The author's early conclusion of dominance based on gene retention rates is not well supported. Based on gene loss, the relative difference is that subgenome A lost 2.16% of duplicated copies and subgenome B lost 2.88% (Supplementary figure 10). Unless the authors can back this up with some meaningful statistics, this should be considered as completely expected under a random process and therefore the authors should remove any mention of dominance in the manuscript based on this evidence alone.

Our response: We agree. Slight difference on gene losses between two subgenomes does not support asymmetric gene losses and subgenome dominance of gene content. We have revised the paragraph and removed the sentences about "gene content dominance" and "asymmetric gene losses".

11. P12L360: Could the authors verify that BAC-end sequences span the breakpoints in the two subgenomes and that they can therefore exclude any assembly errors, that could be an alternative explanation for the rearrangements?

Our response: Very good suggestion. We collected a total of 34,932 mate-paired BAC-end sequences (BES) and mapped 26,350 BES pairs onto 50 chromosomes of common carp. Of the mapped 26,350 BES pairs, only 3,269 were successfully mapped onto two homoeologous chromosomes simultaneously, indicating that two subgenomes are highly divergent with relatively low sequence similarity. We therefore used these 3,269 BES pairs to validate homoeologous chromosomal rearrangement. We identified 96 BES pairs that mapped on the breakpoints of 35 rearrangement regions between two subgenomes, accounting to 38.04% of the identified 92 rearrangement regions. However, we still could not say that the remaining 57 rearrangement regions were due to assembly errors because the sequence similarity between two subgenomes is too low, as shown by the low homoeologous mapping ratio, and the BES pairs may not be able to map onto the breakpoints of rearrangement regions in two homoeologous chromosomes simultaneously. We have added the result about the BES validation in the context. Supplementary table 26 was also added accordingly to support the result.

12. P13L379: the authors indicate mean Ka/Ks = 0.20 but according to Supp Table 26 this should be "median Ka/Ks". Is this still based on the 10,274 pairs and triplets of genes with *C. idella* as outgroup? Please clarify.

Our response: Thanks for reminder. We checked and found that we used older version data in Supplementary Table. We have replaced the table with correct version of data. Now the manuscript and supplementary table are consistent. The Ka/Ks was calculated based on 12,844 conserved protein-coding genes between common carp and zebrafish using codeml model implemented in PAML packages. We have added the details into Methods for clarifying.

13. P13: The authors do not know precisely which clade the donor of subgenome A originates from. It may be evolutionarily more distant to *C. idella* than the donor of subgenome B. The authors should discuss the implications of these differences for evolutionary rate estimations. If population sizes, generation times, etc imposed a different molecular clock on the donor of subgenome A prior to the allotetraploidisation, then surely this could conceivably generate differences in Ka/Ks ratio (even if theoretically it should not) between the two subgenomes when measured today. It would be good if the authors could think of a control to rule out that this is the reason behind their observation. For example, comparing TE evolutionary rates might be useful in this regard.

Our response: We agree with the comment. Consistent evolutionary rate (nucleotide substitution rate) of two subgenomes (diploid progenitors) prior to the allotetraploidisation is fundamental basis for investigating subgenome origins and evolutionary history. It is very good suggestion to compare TE evolutionary rates between two subgenomes. As shown in Figure 2e, TE sequence divergence between two subgenomes displays high degree of overlap, suggesting TE evolutionary rate indeed highly similar between two subgenomes (diploid

progenitors). It would further support the evolutionary rate consistency at genome level of two subgenomes. We have added sentences into the context for brief discussion.

14. P14L414. I only see 6 clusters in Supplementary Figure 13, not 8.

Our response: Thanks for the reminder. We double-checked the paragraph and supplementary figures. The 8 clusters were demonstrated in Supplementary Figure 10 (previous Supplementary Figure 12), not in Supplementary Figure 12 (previous Supplementary Figure 13). We have revised the related paragraph of the manuscript to avoid potential confusion.

15. P14L415: I do not understand what the authors mean by “assigned to different co-expression clusters” nor later with “spatial partitioning”.

Our response: We built co-expression clusters across all 12 tissues using Pearson’s correlation and Ward’s method in the R function `hclust`, and visualized using the R function `heatmap` (`ggplot2`). When two homoeologous genes were assigned to the same co-expression cluster, they were considered that have similar spatial expression patterns in 12 tissues. On the other hand, two homoeologous genes assigned to different co-expression clusters indicate that they have different expression patterns in 12 tissues (spatial expression partitioning).

16. Figure 3d: Please indicate how many genes are comprised in the 3 clusters. How do these clusters relate to those in Supp Figure 13?

Our response: We further built co-expression clusters in 8,291 homoeologous gene pairs using expression values in 12 tissues of diploid grass carp *C. idella* as outgroup. We identified three distinct sub-clusters including: 306 orthologous triplets were differentially expressed in *C. idella* and in two subgenomes of *C. carpio* (sub-cluster I); 228 orthologous triplets that have similar co-expression patterns in grass carp genome and subgenome B of common carp but different co-expression patterns in subgenome A (sub-cluster II); 293 orthologous triplets that have similar co-expression patterns in grass carp genome and subgenome A of common carp but different co-expression patterns in subgenome B (sub-cluster III). These three clusters indicate potential homoeologous subfunctionalisation in allotetraploid genome of *C. carpio*. Supplementary Figure 13 (Supplementary Figure 12 in the revised supplementary file) demonstrated co-expression patterns of 8,291 homoeologous gene pairs in 12 tissues without diploid outgroup, which was used to identify a total of 1,986 homoeologous gene pairs with spatial expression divergence. Both figures illustrated results from different approaches. Majority of the genes of three sub-clusters in Figure 3d were included in 1,986 homoeologous genes. The Venn diagram have been added as Supplementary Figure 11 to illustrate the result.

17. P15L441 and Figure 4c. The legend of Figure 4 says that the two highly divergent clusters were defined on the basis of one of the two homeologs being “completely silenced in all 12 tissues.” When looking at Figure 4c this appears not to be the case, some genes in the upper right panel (extinguished in B) are actually expressed, some very highly so. Please explain this experiment more rigorously.

Our response: Indeed, some genes were still transcribed as shown in the upper right panel. The co-expression clusters were built based the expression values of homoeologous gene pairs, which may not be necessarily restricted as completely silenced in all 12 surveyed tissues. We have removed the inaccurate statement such as

"silenced" in both main text and figure legend, and replaced with "suppressed". We also revised the section in the result with brief introduction of the experiment.

18. Figure 4: please use different colour scheme than red/blue for everything throughout the figure. Blue means B in 4e but A in 4b/d and methylation in 4a and low expression in 4c. It does not help with clarity.

Our response: We have changed colour scheme as suggested.

19. P15L445: The authors must mean Figure 4e

Our response: We have corrected it.

20. P18L542: I do not understand Figure 4a. How can the authors conclude that methylation change was "faster"? The authors seem to compare the absolute ratio of synonymous changes (Ks) with a percentage of methylation change. This does not make immediate sense to me because while Ks is indeed an approximation of evolutionary time, the % methylation is definitely not (there is no indication that it varies at a constant rate with time). Could the authors clarify this? Also, the legend mentions arrows for peak values, but I cannot see them.

Our response: We adapted the method from recent paper published on *Genome Biology* by Jeffrey Chen's lab, entitled "Epigenomic and functional analyses reveal roles of epialleles in the loss of photoperiod sensitivity during domestication of allotetraploid cottons (<https://doi.org/10.1186/s13059-017-1229-8>)". Jeffrey Chen's lab focuses on genomics and epigenetics of polyploidy and heterosis for decades and published many papers on epigenomics and evolution of various allotetraploid species. Both common carp and domesticated cotton are allotetraploid and may experience similar epigenetic regulation mechanisms, therefore, we follow their method adapted from Takuno *et al.* (2011) to explore potential methylation regulations in allotetraploid carp genome. As shown in Figure 4a, our result indicated that CG methylation change rate is higher than the neutral sequence substitution rate of the coding sequences in surveyed homoeologous genes. We have added a paragraph in the Methods and cited related literatures. We have also added arrows in the figure as legend mentions and added the label of the axes.

References:

Song Q, Zhang T, Stelly DM, Chen ZJ. Epigenomic and functional analyses reveal roles of epialleles in the loss of photoperiod sensitivity during domestication of allotetraploid cottons. *Genome biology*. 2017 Dec;18(1):99.
Takuno S, Gaut BS. Body-methylated genes in *Arabidopsis thaliana* are functionally important and evolve slowly. *Molecular Biology and Evolution*. 2011 Aug 2;29(1):219-27.

21. P19L556: what test was performed to compute this p value?

Our response: We performed Student's T test to compute the p value. We have added "Student's T test" before the p value to clarify.

22. P19L556-560: repetition of the previous sentence.

Our response: We have carefully checked the sentences. The mentioned sentence describes the CG methylation difference in the promoter regions of the highly conserved 8,291 homoeologous gene pairs in two subgenome, while the previous sentence describes the CG methylation difference based on all annotated genes in two subgenomes. Both results refer to Figure 4b. We have re-phrased the sentences to prevent potential confusion.

23. P20L586: please annotate better Figure 4f: what is the horizontal arrow with vertical bars, what are the black bars in the *eif3ia/b* diagrams? The text mentions two additional genes that are absent for the figure.

Our response: The horizontal arrow with vertical bars indicates the gene structure and orientation of *eif3i*. The black bars in the *eif3ia/b* diagrams indicate the CG methylation sites. We have improved Figure 4f as suggested, including adding annotations of "5'→3'" and "CG methylation", changing colour. We have revised the sentence and remove those two genes, which were previously selected as examples in the early version of the manuscript.

24. P22 and Figure 5a and 5c: the rationale for selecting these two genes based on the data is not clear.

Our response: We have removed this session and Figure 5 as reviewer's suggestion. We will perform gene function verification experiments (i.e. using CRISPR techniques) of both *bco2* and *oca2* genes in carp. We will submit another article when we collect more convincing evidences.

25. P22: what is the expression of *bco2a-1* in control (i.e. grey) *C. carpio* strains? This data should be provided to conclude that this gene is causative.

Our response: Same as above.

26. I believe that homoelogenous/homoelogs is misspelled throughout the manuscript (it should be homeologous).

Our response: We previously also confused about two different spellings on "homoelogenous" vs. "homeologous", both have been used in previous literatures about allotetraploid, until we read the review article "Homoeologs: What Are They and How Do We Infer Them?" (<https://doi.org/10.1016/j.tplants.2016.02.005>). This review discussed usages of both "homoelogenous" and "homeologous", and indicated that "homoeolog" and its derivatives are more common spelling and have significantly more mentions than "homeolog". Besides, the latest published allotetraploid genome of *Xenopus laevis* (<https://doi.org/10.1038/nature19840>) also used the spelling of "homoeologous" to describe similar chromosomes (genes) from distinct progenitors. Considering that both *Xenopus laevis* and *Cyprinus carpio* are very typical allotetraploid vertebrates, and the readers of both articles may have great level of overlapping, we decided to follow the spelling "homoeologous" used in the *Xenopus* paper.

Reviewers' Comments:

Reviewer #1:

Remarks to the Author:

Response to the revised version of Ms197979

The authors have dealt adequately with the reviewer criticisms of the submitted version. Some minor comments are below.

Regarding the authors responses to my specific comments and questions –

1. Fig 1 and 2 are OK.
2. Page 3, line 70 – 'polyploidisation' is misspelt. Change to 'polyploidisation'.

3. Fig 2a. The authors address my comments (and misunderstandings) adequately.

Fig 3c, e and f. Corrections are OK.

Fig 4 legend. Can I just check that the text for 4d is accurate? As I see it, this fig shows the methylation pattern of genes included in Cluster's I and II of Fig 4c. The text indicates that this relates instead to Fig 4a. This needs checking.

Line 501 – Student's t test. Im not sure the t is capitalised.

Text – whilst the past tense is correctly employed throughout, there are a few occasions when the present tense was inappropriately used. For example, line 418-421 discusses novel findings in the present tense ('is') when most of the text is in the past tense. Others include lines 235, 273, 408, 409, 414, 419-22, 488, and 552. Sorry, if this is being pedantic.

This version omits the Hebao red carp story which I think is appropriate, and help the focus on the main achievement of this work. It also omits the reference to geo-climatic phenomena, as suggested.

Andrew R. Cossins

Reviewer #2:

Remarks to the Author:

I find that on the whole the authors did a good job with addressing the issues raised. My remaining requests are mainly for clarification. The most important is the following:

1. Page 17 and Figure 4a. I appreciate the author's response that they used a well-established method inspired by Jeffrey Chen's work in polyploidy plants, but the vertebrate (fish) genomics community may not be familiar with it. Therefore the rationale behind the DmCG vs Ks comparison should be explained in more details. The most important is to explicit what is being computed in Figure 4a for DmCG for the CG methylation change. It must be explained in the methods section. Please indicate how many Cytosine sites were used and how these were identified. In the corresponding section P17, please cite reference 44 (Song et al. 2017). Also, do I understand well that Ks here measures the rate of silent substitution since the two subgenome split 23 Mya, while methylation (which is reset genome wide at every generation) differences reflect a possible dominance established only since the allotetraploidy ~12 Mya. Is this correct? It may be worth clarifying/mentioning this in the text.

I also have a few minor comments:

2. Page 3, line 70: The date of the salmonid WGD: it has been estimated at approximately 100 million years rather than 80 million years (Macqueen and Johnston (2014), Berthelot et al. (2015)).

3. Page 8, line 232 and 233: "Danio-Cyprinus paralogous" and "Cyprinus-Sinocyclocheilus paralogous" should be "orthologous" in both cases, I believe.

4. Page 9, line 251: The current phrasing creates some confusion, I believe that "subgenomes A and B had to be differentiated" should be "subgenomes A and B differentiated".

5. Page 12, lines 340-343: The sentence "... a total of 7,536 expressed homoeologous gene pairs (~91%) had expression differences greater than 2-fold change, including 4,719 and 5,403 homoeologs..." is confusing. Greater than 2-fold change compared to what? And 4,719 + 5,403 genes cannot be 'included' in 7,536 genes. Please clarify.

6. Page 35: in the legend of figure 3d, please indicate the tissues corresponding to the different columns. Ideally this should be the same as the order of tissues in 3c, for the sake of readability.

7. Page 17, lines 484-485. Please define the PCG acronym.

Point-to-Point Response to Reviewers' Comments:

Reviewer #1 (Remarks to the Author):

The authors have dealt adequately with the reviewer criticisms of the submitted version. Some minor comments are below.

Regarding the authors responses to my specific comments and questions –

1. Fig 1 and 2 are OK.
2. Page 3, line 70 – ‘polyploidisation’ is misspelt. Change to ‘polyploidisation’.

Our response: Thanks for the reminder. We have corrected the misspelt word.

3. Fig 2a. The authors address my comments (and misunderstandings) adequately.

Fig 3c, e and f. Corrections are OK.

Fig 4 legend. Can I just check that the text for 4d is accurate? As I see it, this fig shows the methylation pattern of genes included in Cluster’s I and II of Fig 4c. The text indicates that this relates instead to Fig 4a. This needs checking.

Our response: Thanks. Fig 4d indeed shows the methylation pattern of genes included in Cluster’s I and II of Fig 4c. We have corrected the figure legend.

Line 501 – Student’s t test. Im not sure the t is capitalised.

Our response: We have modified as “t-test”.

Text – whilst the past tense is correctly employed throughout, there are a few occasions when the present tense was inappropriately used. For example, line 418-421 discusses novel findings in the present tense (‘is’) when most of the text is in the past tense. Others include lines 235, 273, 408, 409, 414, 419-22, 488, and 552. Sorry, if this is being pedantic.

Our response: We have revised the tenses according to the suggestion.

This version omits the Hebao red carp story which I think is appropriate, and help the focus on the main achievement of this work. It also omits the reference to geo-climatic phenomena, as suggested.

Reviewer #2 (Remarks to the Author):

I find that on the whole the authors did a good job with addressing the issues raised. My remaining requests are mainly for clarification. The most important is the following:

1. Page 17 and Figure 4a. I appreciate the author's response that they used a well-established method inspired by Jeffrey Chen's work in polyploidy plants, but the vertebrate (fish) genomics community may not be familiar with it. Therefore the rationale behind the DmCG vs Ks comparison should be explained in more details. The most important is to explicit what is being computed in Figure 4a for DmCG for the CG methylation change. It must be explained in the methods section. Please indicate how many Cytosine sites were used and how these were identified. In the corresponding section P17, please cite reference 44 (Song et al. 2017). Also, do I understand well that Ks here measures the rate of silent substitution since the two subgenome split 23 Mya, while methylation (which is reset genome wide at every generation) differences reflect a possible dominance established only since the allotetraploidy ~12 Mya. Is this correct? It may be worth clarifying/mentioning this in the text.

Our response: Thanks for the suggestion. We have added more details in the DmCG identifying section in methods. The total cytosine residues at CG sites across the whole genome is 42,911,370 and three biological replicates share a total of 21,275,948 methylated CG sites, which have been also included in the text, and cited the reference of Song, et al. 2017 (now it is reference 42).

The Ks indeed measures the rate of neutral substitution since two subgenome divergence at ~23 Mya. The methylation rate is more complex just as mentioned, methylation is partially reset at every generation. The accumulated methylation differences, however, may still be estimated by comparing the methylation changes of two homoeologous copies, which reflects the methylation trend of two subgenomes at genome level since their divergence at 23 Mya, no matter they were in the independent genomes or in the merged allotetraploid genome. We have revised the sentences in the context to clarify.

I also have a few minor comments:

2. Page 3, line 70: The date of the salmonid WGD: it has been estimated at approximately 100 million years rather than 80 million years (Macqueen and Johnston (2014), Berthelot et al. (2015)).

Our response: Thanks. We have cited the two articles related to the salmonid WGD and modified the time of salmonid WGD as 100 million years.

3. Page 8, line 232 and 233: “Danio-Cyprinus paralogous” and “Cyprinus-Sinocyclocheilus paralogous” should be “orthologous” in both cases, I believe.

Our response: Thanks. We have replaced the “paralogous” with “orthologous”.

4. Page 9, line 251: The current phrasing creates some confusion, I believe that “subgenomes A and B had to be differentiated” should be “subgenomes A and B differentiated”.

Our response: We have removed “had to be” for clarification.

5. Page 12, lines 340-343: The sentence “... a total of 7,536 expressed homoeologous gene pairs (~91%) had expression differences greater than 2-fold change, including 4,719 and 5,403 homoeologs...” is confusing. Greater than 2-fold change compared to what? And 4,719 + 5,403 genes cannot be ‘included’ in 7,536 genes. Please clarify.

Our response: Thanks. The 7,536, 4,719 and 5,403 differently expressed homoeologous gene pairs are the gene pairs have expression differences in at least one tissue of 12 surveyed tissues. The summed number of 4,719 + 5,403 (summed as 10,122) were greater than 7,536 indicated that some homoeologous gene pairs (10,122-7,536=2,586) had swinging expression bias in 12 tissues (e.g. one homoeologous copy may be dominant in partial tissues but suppressed in several other tissues). We have clarified this in the next sentence, read as *"a total of 7,536 expressed homoeologous gene pairs (~91%) had expression differences greater than 2-fold change in at least one tissue, including 4,719 and 5,403 homoeologs having higher expression in subgenomes A and B, respectively, of which, 2,133 and 2,817 homoeologs had higher expression values exclusively in the 12 tissues of subgenomes A and B, respectively, while 2,586 homoeologs had swinging expression bias in 12 tissues"*.

6. Page 35: in the legend of figure 3d, please indicate the tissues corresponding to the different columns. Ideally this should be the same as the order of tissues in 3c, for the sake of readability.

Our response: Thanks for the suggestion. We have reordered the tissues in figure 3c. Now they are in the same order as 3d.

7. Page 17, lines 484-485. Please define the *PCG* acronym.

Our response: Thanks. *PCG* is a proxy *P* value of DNA methylation level and the criteria of $PCG < 0.05$ is used to define CG body-methylated genes. We have defined this in the methods section.